# Optimal Stopping in Latent Diffusion Models

Yu-Han Wu [1 2]  Quentin Berthet [2]  Gérard Biau [1]  Claire Boyer [3]  Romuald Elie [2]  Pierre Marion [4]

## Abstract

We identify and analyze a surprising phenomenon of *Latent* Diffusion Models (LDMs) where the final steps of the diffusion can *degrade* sample quality. In contrast to conventional arguments that justify early stopping for numerical stability, this phenomenon is intrinsic to the dimensionality reduction in LDMs. We provide a principled explanation by analyzing the interaction between latent dimension and stopping time. Under a Gaussian framework with linear autoencoders, we characterize the conditions under which early stopping is needed to minimize the distance between generated and target distributions. More precisely, we show that lower-dimensional representations benefit from earlier termination, whereas higher-dimensional latent spaces require later stopping time. We further establish that the latent dimension interplays with other hyperparameters of the problem such as constraints in the parameters of score matching. Crucially, this framework suggests that the reconstruction quality of the autoencoder alone can serve as a proxy to estimate the potential performance of the full LDM. Experiments on synthetic and real datasets illustrate these properties, underlining that early stopping can improve generative quality. Together, our results offer a theoretical foundation for understanding how the latent dimension influences the sample quality, and highlight stopping time as a key hyperparameter in LDMs.

## 1. Introduction

A pivotal advancement in the evolution of diffusion models is the introduction of the Latent Diffusion Model (LDM,

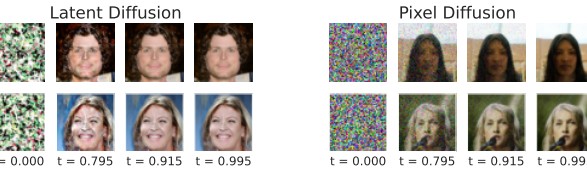

*Figure 1.* Samples generated with a latent diffusion model (LDM) and a pixel-space diffusion. In the LDM, the before-last sample is nearly denoised and indistinguishable from the final one, whereas in the pixel-space model stronger noise remains at that timestep. See Appendix E for more examples.

Rombach et al., 2022). Instead of performing the computationally intensive diffusion process in the high-dimensional pixel space, LDMs first compress the data into a lower-dimensional latent space using a pretrained autoencoder (AE, Kingma & Welling, 2013). The diffusion steps then occur within this more manageable latent representation, significantly reducing computational requirements and training time without a meaningful loss of quality. The standard training pipeline for LDMs operates as a two-stage process: an AE is trained first to compress the data, followed by the training of a diffusion model within the resulting latent space. While recent research has focused on optimizing AE training to yield better latent representations for diffusion (e.g., Kouzelis et al., 2025; Zhou et al., 2025), selecting the appropriate latent dimension remains a critical challenge. The dimensionality must be fixed during the initial AE training phase, yet identifying the dimension that yields the optimal tradeoff between generation quality and compute cost typically requires the resource-intensive process of training a subsequent LDM for each candidate. Furthermore, distinct behavioral differences exist between pixel-space and latent-space diffusion. As shown in Figure 1, latent-generated images appear to visibly stabilize in the final diffusion steps, in contrast to pixel-space diffusion, where the final steps are crucial for removing remaining noise. This challenges the common assumption that LDMs yield the best samples at the final timestep, suggesting instead that early stopping may actually improve image quality, as the decoder may introduce high-frequency artifacts in the final stages.

**Contributions and organization.**  Our goal in this paper is to investigate the optimal stopping time and latent dimensionality for LDMs. More specifically, our contributions are

[1]LPSM, Sorbonne Université [2]Google DeepMind [3]LMO, Université Paris-Saclay [4]Inria, École Normale Supérieure, PSL Research University. Correspondence to: Yu-Han Wu <yhwu@google.com>, Pierre Marion <pierre.marion@inria.fr>.

*Proceedings of the $43^{rd}$ International Conference on Machine Learning*, Seoul, South Korea. PMLR 306, 2026. Copyright 2026 by the author(s).

as follows:

- Working within a Gaussian framework, we explicitly derive the Wasserstein-2 distance, which is equivalent to the Fréchet distance in the Gaussian setting, between the true data distribution and the backward diffusion. In particular, we highlight that the diffusion process is identical to injecting noise into the latent representations prior to decoding (Section 3).

- We determine the optimal latent dimension and stopping time for diagonal and general covariances. In particular, for data on a linear subspace, we prove that projecting onto that subspace is optimal. This reveals a trade-off: early generation steps favor lower dimensions, while later steps require higher dimensions for faithful reconstruction (Section 4).

- When the score is learned by a restricted class of parametrized models (i.e., when the weights of the model are capped), we further establish the existence of an optimal latent projection that optimizes the backward diffusion, and we investigate how its dimension depends on the model class constraint and the data covariance (Section 5).

- Our theoretical analysis suggests the existence of an optimal interval for each LDM, which can be identified solely by examining the corresponding noisy AEs. Building on this insight, we first validate our predictions on GMM data with exact scores and on MNIST with linear autoencoders, then demonstrate that these results also have predictive power in practical settings, i.e., we are able to predict the optimal stopping time and optimal interval for different LDMs by the FID curves of the corresponding version of noisy AEs. We train a diverse set of AEs and their corresponding LDMs, ensuring a controlled comparison by maintaining a consistent total parameter count across models. We then show that the Fréchet inception distance (FID) of the decoded noisy latent representations closely tracks that of the backward latent diffusion process. Notably, the FID curve of different LDMs cross at the same time as their corresponding version of noisy AEs (Section 6).

## 2. Related work

**Learning low-dimensional data with diffusion models.** Riemannian Diffusion Models, introduced by Huang et al. (2022); De Bortoli et al. (2022), generalize the diffusion process to operate on Riemannian manifolds and preserve a known geometric structure by design. Subsequent theoretical work has analyzed the behavior of standard Denoising Diffusion Probabilistic Models (DDPMs) under the manifold hypothesis, demonstrating that they can implicitly adapt to the data's intrinsic dimension without explicit

knowledge of the manifold (Tang & Yang, 2024; George et al., 2025). Further improvements in computational and memory efficiency were introduced by LDMs (Rombach et al., 2022) by first training a compression model to transform images into a lower-dimensional latent space, from which the original data can be reconstructed at high fidelity. In practice, this approach is implemented with a regularized VAE (Esser et al., 2021). The LDM is then trained in the latent space. Building on this core concept, LDMs have been extended to new domains, such as the generation of high-resolution videos (Blattmann et al., 2023). Furthermore, extensive research has focused on improving LDM's sampling quality, including methods like aligning encoded images with DINOv2 representations (Yu et al., 2024), and enhancing the robustness of the latent space through explicit or implicit equivariance constraints (Kouzelis et al., 2025; Skorokhodov et al., 2025; Zhou et al., 2025). In contrast to standard diffusion models, theoretical properties of LDMs have been little studied; in this work, we investigate the connection of the latent dimension with diffusion stopping time and score matching regularization.

**Optimal stopping time of diffusion models.** Focusing on a theoretical analysis of this phenomenon, Achilli et al. (2025) investigate the optimal stopping time for diffusion models under the assumption that the data is concentrated on a low-dimensional manifold, a concept formalized by the Hidden Manifold Model (Goldt et al., 2020). Closer to our contribution is the work of Hurault et al. (2025). They also investigate the scenario where the true data distribution is Gaussian. Their analysis focuses on learning the score function using SGD, and allows them to determine an optimal stopping time. However, the study of these authors is limited to the diffusion model and did not consider the two-stage architecture of LDMs. Furthermore, the relationship between the data dimension and the derived optimal stopping time remained unexplored in their findings. In contrast, our work directly investigates the influence of the latent dimension on the optimal stopping time by incorporating an autoencoder into the diffusion model framework. We also demonstrate the need of early stopping without discretization of the backward diffusion process. Finally, recent findings suggest that full decoding (at $t = 0$) does not always yield the lowest FID (e.g., Jayasumana et al., 2024). While this previous work attributed this behavior to be an artifact of the FID metric, in this work we challenge the common assumption that LDMs yield the best samples at the final timestep, suggesting instead that early stopping may actually improve image quality.

## 3. Notations and problem setup

This section introduces the mathematical formalism of diffusion models in the considered setting.

**Latent Diffusion Models.** Let $p_0$ be an unknown distribution in $\mathbb{R}^D$. With a slight abuse of notation, we use in the following the same notation for a distribution and its density function. The goal of diffusion models is to generate new observations following $p_0$, given an i.i.d. sample $(X_1, \ldots, X_n)$ drawn from $p_0$. The mechanism is as follows. Given a final diffusion time $T > 0$, a latent dimension $d \leq D$, a semi-orthogonal projection matrix $P \in \mathbb{R}^{d \times D}$, and a scalar function $w : [0, T] \to \mathbb{R}$, the latent forward variance-preserving (VP)-SDE (Song et al., 2020) is defined by

$$dP\overrightarrow{X_t} = -w_t^2 P\overrightarrow{X_t}dt + \sqrt{2w_t^2}dP\overrightarrow{W_t}, \quad P\overrightarrow{X_0} \sim P_{\#}p_0, \tag{1}$$

where $\overrightarrow{W_t}$ is a standard $D$-dimensional Brownian motion. The role of the matrix $P$ is to perform linear dimension reduction. Two special cases are of interest: first, if $d = D$ and $P$ is the identity matrix, we recover the standard formulation of diffusion models. Second, if $P$ projects on the first few principal components of the sample covariance matrix, this amounts to performing principal component analysis (PCA, Jolliffe, 2002). This projection is equivalent to linear autoencoders (Plaut, 2018), and there exists a pseudo-inverse $P^+ \in \mathbb{R}^{D \times d}$ which allows us to map sample back to $\mathbb{R}^D$. We leave to future work to extend the analysis of this phenomenon to other noise schedules such as EDMs (Karras et al., 2022). Letting $s_P$ be the score function of $P\overrightarrow{X_t}$, i.e., $s_P(x, t) = \nabla \log p_P(x, t)$ where $p_P(\cdot, t)$ is the density function of $P\overrightarrow{X_t}$, the forward diffusion can be reversed in time using the backward process

$$dP\overleftarrow{X_t} = (w_{T-t}^2 P\overleftarrow{X_t} + 2w_{T-t}^2 s_P(P\overleftarrow{X_t}, T - t))dt + \sqrt{2w_{T-t}^2}dP\overleftarrow{W_t}, \quad P\overleftarrow{X_0} \sim P_{\#}p_T, \tag{2}$$

where $\overleftarrow{W_t}$ is a standard $D$-dimensional Brownian motion. This means that the marginal distribution of $P\overleftarrow{X}_{T-t}$ matches the marginal distribution of $P\overrightarrow{X_t}$ (Anderson, 1982). Hence running the backward diffusion allows to generate a sample from $\overleftarrow{X}_T \sim P_{\#}p_0$, and then the pseudo-inverse $P^+$ can be used to map the generated sample back to $\mathbb{R}^D$. Importantly, this procedure requires knowledge of $s_P$, which can be estimated using the training sample.

**Problem setup.** We assume that the data distribution $p_0$ is a centered $D$-dimensional Gaussian with independent components, i.e.,

$$p_0 = \mathcal{N}(0, \Sigma) \tag{3}$$

This specific setting simplifies our study but shall still provide important insights for more general distributions. We consider a hierarchy of latent spaces with increasing dimension $d$ from 1 to $D$. This is formalized by defining the

matrix $P$ in the VP-SDE (1) as the semi-orthogonal projection matrix $P_d O^\top$ where $P_d$ is the projection matrix onto the first $d$ dimensions and $O$ an orthonormal matrix. In this Gaussian framework, the marginal distributions of the backward process are Gaussian with an explicit covariance matrix:

**Lemma 3.1.** *The distribution of $P_d O^\top \overleftarrow{X}_{T-t}$ is*

$$P_d O^\top \overleftarrow{X}_{T-t} \stackrel{d}{=} a_t P_d O^\top \overrightarrow{X_0} + b_t Z \tag{4}$$
$$\sim \mathcal{N}(0, a_t^2 I_d + b_t^2 P_d O^\top \Sigma O P_d^\top),$$

*where $Z \sim \mathcal{N}(0, I_d), a_t = \sqrt{1 - b_t^2}$ and $b_t = e^{-\int_0^t w_t^2 dt}$, and its score is*

$$s_{P_d}(x, t) = -(a_t^2 I_d + b_t^2 P_d \Sigma P_d^\top)^{-1}x, \quad x \in \mathbb{R}^d.$$

Here, the pseudo-inverse of $P$ is $P^+ = OP_d^\top$, and the decoded backward distribution in the original data space is therefore given by $Q\overleftarrow{X}_{T-t} \sim p_{d,t,O}(\Sigma)$, where $Q = OP_d^\top P_d O^\top$ and

$$p_{d,t,O}(\Sigma) := \mathcal{N}(0, a_t^2 Q + b_t^2 Q\Sigma Q).$$

In practical scenarios, the true covariance matrix $\Sigma$ is unknown and must be estimated from data. The derivation above only depends on $p_0$ through its covariance matrix $\Sigma$. Thus, by estimating the covariance matrix with $\hat{\Sigma}$, the estimated backward diffusion process follows a Gaussian distribution given by (see Appendix A for a formal discussion)

$$p_{d,t,O}(\hat{\Sigma}) = \mathcal{N}(0, a_t^2 Q + b_t^2 Q\hat{\Sigma}Q).$$

Crucially, the expression (4) together with the pseudo-inverse $P^+$ reveal that the generative process can be reinterpreted as a "noisy autoencoder": a model that encodes the data, injects noise into the latent representation, and decodes it. Our goal is to quantify the discrepancy between the target distribution $p_0$ and the decoded backward distribution $p_{d,t,O}(\Sigma)$ as a function of the latent dimension $d$ and time $t$. To this end, we utilize the Wasserstein-2 distance (Villani, 2008), which corresponds to the Fréchet distance in the Gaussian setting (Heusel et al., 2017):

$$d_F^2(\mathcal{N}(\mu_1, \Sigma_1), \mathcal{N}(\mu_2, \Sigma_2)) = \|\mu_1 - \mu_2\|_2^2 + \text{tr}(\Sigma_1 + \Sigma_2 - 2(\Sigma_2^{1/2}\Sigma_1\Sigma_2^{1/2})^{1/2}). \tag{5}$$

We define our primary quantity of interest as:

$$\Delta_{d,t,O}(\Sigma) := d_F^2(p_0, p_{d,T-t}(\Sigma)).$$

Whenever $O = I_D$ we simply denote it by $\Delta_{d,t}(\Sigma) := \Delta_{d,t,I_D}(\Sigma)$. We note that $\Delta_{d,t,O}(\Sigma)$ is defined for $t \in [0, T]$ and $d \in \{1, \ldots, D\}$. We will also be interested in the plug-in estimator $\Delta_{d,t}(\hat{\Sigma}) = d_F^2(p_0, p_{d,T-t}(\hat{\Sigma}))$.

# 4. Optimal dimension reduction and stopping time

In this section, we address the important question of how dimensionality reduction affects the diffusion process with respect to the intrinsic geometric structure of the data. Our analysis focuses on selecting the rank $d$ of the projection matrix $P_d$ and the stopping time of the diffusions, by examining the dependence of $\Delta_{d,t,O}(\Sigma)$ and $\Delta_{d,t,O}(\hat{\Sigma})$ on $d$ and $t$. We begin by the simpler case where the covariance matrix is assumed to be diagonal before moving on to a general covariance matrix (with unknown eigenvectors).

## 4.1. Non-monotonicity of the Fréchet distance

In this subsection and the next one, we focus on the scenario where $p_0$ has independent components, i.e. $\Sigma = \mathrm{diag}(\sigma_1^2, \ldots, \sigma_D^2)$, and diagonal estimators $\hat{\Sigma} = \mathrm{diag}(\hat{\sigma}_1^2, \ldots, \hat{\sigma}_D^2)$, where the diagonal elements are the sample variances $\sigma_k^2 = \frac{1}{n} \sum_{i=1}^{n} X_{ik}^2$. We assume that $n$ is sufficiently large such that $\hat{\sigma}_1^2 \geq \ldots \geq \hat{\sigma}_D^2 > 0$ holds with high probability. In addition, we consider $O = I_D$. This subsection examines the non-monotonic behavior of the Fréchet distance as a function of diffusion timesteps, challenging the intuitive expectation of monotonic evolution. The common belief of monotonicity (Jayasumana et al., 2024) implies that a stopping time closer to $T$ consistently yields a smaller Fréchet distance. First, we derive a necessary and sufficient condition for this non-monotonicity to occur in the scenario where the target distribution is Gaussian, as in (3).

**Proposition 4.1.** *For $d \in \{1, \ldots, D\}$, the Fréchet distance $\Delta_{d,t}(\Sigma)$ is non-increasing with respect to $t$. On the other hand, $\Delta_{d,t}(\hat{\Sigma})$ is non-increasing if and only if*

$$\sum_{d'=1}^{d} (1 - \frac{\sigma_{d'}}{\hat{\sigma}_{d'}})(1 - \hat{\sigma}_{d'}^2) \geq 0. \tag{6}$$

Roughly speaking, the variance of each generated component interpolates monotonically from an initial value close to 1 (the standard normal prior) toward the estimated variance $\hat{\sigma}_d^2$ as $t$ increases from 0 to $T$. When the true variance $\hat{\sigma}_d^2$ lies between these two endpoints, the process variance must pass through $\hat{\sigma}_d^2$ at some intermediate time, at which point the Fréchet distance is minimized. Condition (6) captures precisely when these intermediate crossings dominate, making early stopping beneficial. The distance $\Delta_{d,t}(\hat{\Sigma})$ is therefore minimized when the process variance is closest to the true set of variances $(\sigma_{d'}^2)_{1 \leq d' \leq d}$, which happens before time $T$ under condition (6). For a clearer understanding, consider the scenario where $p_0$ is a distribution lying in a linear subspace that is isomorphically equivalent to $\mathbb{R}^{d_0}$. In other words, suppose that $\sigma_D = \ldots = \sigma_{d_0+1} = 0$, and also $\hat{\sigma}_D = \ldots = \hat{\sigma}_{d_0+1} = 0$. Let us first consider the case where there is no projection, i.e., $d = D$. Then, the left-hand side

of (6) can be rewritten

$$\sum_{d'=1}^{D} (1 - \frac{\sigma_{d'}}{\hat{\sigma}_{d'}})(1 - \hat{\sigma}_{d'}^2) = \sum_{d'=1}^{d_0} (1 - \frac{\sigma_{d'}}{\hat{\sigma}_{d'}})(1 - \hat{\sigma}_{d'}^2) + D - d_0.$$

For large enough $n$ and with high probability, $|\sigma_{d'} - \hat{\sigma}_{d'}| \leq \hat{\sigma}_{d'}$ for every $d' \in \{1, \ldots, d_0\}$, we may deduce that

$$\sum_{d'=1}^{D} (1 - \frac{\sigma_{d'}}{\hat{\sigma}_{d'}})(1 - \hat{\sigma}_{d'}^2) \geq D - \left(2 + \max_{d' \in \{1, \ldots, d_0\}} \hat{\sigma}_{d'}^2\right) d_0.$$

The last term is positive as long as the ambient dimension $D$ is large enough. Therefore, in this context, $d_F(\overleftarrow{X}_t, \overrightarrow{X}_0)$ is non-increasing. However, if projecting the diffusion onto the $d_0$-dimensional linear subspace in which the data distribution lies, the $D - d_0$ term in the computation above vanishes, and we are left with the sum up to $d_0$. Then the behavior of the Fréchet distance is linked to how the model estimates the variances of the data. If, for most $d'$, the sign of $1 - \sigma_{d'}/\hat{\sigma}_{d'}$ matches the sign of $1 - \hat{\sigma}_{d'}^2$, the Fréchet distance exhibits monotonic behavior. Conversely, if most of the signs differ, the Fréchet distance is non-monotonic. In addition, inspecting the inequality above, we observe that non-monotonicity is more likely to occur when the latent dimension $d_0$ is small. This insight suggests that early stopping can improve the backward diffusion process, bringing the generated distribution closer to the data distribution. We next ask the reverse question: given a stopping time $t$, what is the optimal latent dimension?

## 4.2. Optimal projection at time $t$

In this subsection, we continue the study of the interaction of the dimension of projection and the stopping time. In contrast to the previous subsection, we show that for each fixed time $t$, there exists an optimal projection $P_d$. We still consider Gaussian data with independent components (3). Recall that $a$ is defined in (10) and that it is an increasing map from $[0, T]$ to $[0, a_T]$. We then let $\bar{a}^{-2} : \mathbb{R} \cup \{\infty\} \rightarrow [0, T]$ be the extended inverse function of $a^2$ (see plot in Figure 2), meaning that

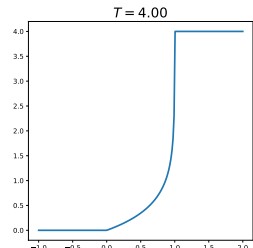

*Figure 2.* $\bar{a}^{-2}$ in the Ornstein-Uhlenbeck process.

$$\bar{a}^{-2}(x) = \begin{cases} 0, & \text{for } x < 0, \\ a^{-2}(x), & \text{for } x \in [0, a_T^2], \\ T, & \text{for } x \in (a_T^2, \infty]. \end{cases} \tag{7}$$

In particular, for $t \in [0, T]$, $\bar{a}^{-2}(a_t^2) = t$. For $d \in$

$\{2, \ldots, D\}$, we then let

$$t_d = T - \bar{a}^{-2}\Big(\frac{3\sigma_d^2}{(1 - \sigma_d^2)_+}\Big)$$

and

$$\hat{t}_d = T - \bar{a}^{-2}\Big(\frac{4\sigma_d^2 - \hat{\sigma}_d^2}{(1 - \hat{\sigma}_d^2)_+}\Big).$$

By convention, we let $\hat{t}_1 = t_1 = 0$ and $\hat{t}_{D+1} = t_{D+1} = T$. Observe that the times $t_d$ are in increasing order and between $0$ and $T$. Given these time partitions, we can characterize the optimal projection dimension, both for the exact backward process and the one incorporating score estimation, with the aim of minimizing the distance between the generated and target distributions.

**Proposition 4.2.** *Assume that $0 < \sigma_D < \cdots < \sigma_1$. Then, for $d \in \{1, \ldots, D\}$ and $t \in [t_d, t_{d+1})$,*

$$d = \underset{d' \in \{1, \ldots, D\}}{\arg\min} \Delta_{d', t}(\Sigma)$$

*Furthermore, with high probability, the $\hat{\sigma}_d$ and the $\hat{t}_d$ are well-ordered. In this case, for $t \in [\hat{t}_d, \hat{t}_{d+1})$*

$$d = \underset{d' \in \{1, \ldots, D\}}{\arg\min} \Delta_{d', t}(\hat{\Sigma})$$

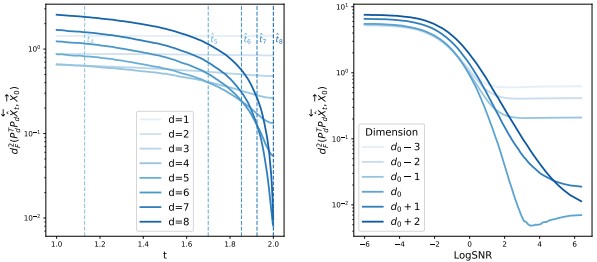

*Figure 3.* Plots of $\Delta_{d,t}(\hat{\Sigma})$ as a function of the diffusion time $t$, for two sets of variances. (left) All the $\sigma_i$ are nonzero. As expected from Proposition 4.2, the $d$-dimensional projection is optimal in $[t_d, t_{d+1})$. (right) The data is supported on a linear subspace of dimension $d_0 = 4$ with $D = 6$. As expected from Proposition 4.3, we observe that the minimum distance is achieved in dimension $d_0$ and with early stopping. LogSNR in the $x$-axis is a remapping of time $t$, defined as $\log(\bar{b}_t^2 / \bar{a}_t^2)$, which we use to increase readability. Experimental details are in Appendix E.

Proposition 4.2 quantifies a direct link between early stopping in the backward diffusion process and dimensionality reduction. It reveals a time-dependent trade-off: distributions at early stages of the backward process are best approximated in lower-dimensional spaces, while higher dimensions become necessary to faithfully reconstruct the data as $t \to T$, as illustrated in Figure 3 (left). In other words, at an early time step, projecting onto an unnecessarily high-dimensional space can introduce more noise than signal, making a lower-dimensional representation more

accurate. Notice that when $4\sigma_d^2 \geq 1$, both $t_d$ and $\hat{t}_d$ are equal to $0$. This implies that a component whose variance is sufficiently large should always be included in the projection, aligning with the intuition that major components are essential for representation. These results hold for backward processes using scores based on either the true or empirical variances. We next characterize the behavior of the optimal latent dimension and stopping time when data lies on a $d_0$-dimensional subspace, providing a similar result to Proposition 4.2. This analysis allows us to precisely determine these two key parameters, as shown next.

**Proposition 4.3.** *Let $\Sigma = \mathrm{diag}(\sigma^2, \ldots, \sigma^2, 0, \ldots, 0)$ with the last $D - d_0$ entries equal to $0$. Let $\varepsilon \in (0, 1)$. Then, there exists $\hat{\delta}_{d_0} \in [0, T]$ such that with probability $1 - 2d_0 e^{-\frac{n}{8}}$,*

$$(d_0, T - \hat{\delta}_{d_0}) = \underset{(d', t) \in \{1, \ldots, D\} \times [0, T]}{\arg\min} \Delta_{d', t}(\hat{\Sigma}).$$

The proposition shows that the optimal generation strategy for data with a low-rank structure involves both early stopping and projection (see Figure 3 (right) for an illustration). The proof indicates that, under the non-monotonicity condition of Proposition 4.1, the optimal early stopping time $T - \hat{\delta}_{d_0}$ is strictly before $T$. Beyond preventing numerical instability as $t \to T$ (e.g., Yang et al., 2023), Proposition 4.3 thus offers a new justification for early stopping. In other words, stopping at a specific time $\hat{\delta}_{d_0}$ is not merely a practical fix, but an optimal strategy to improve generation quality by minimizing the distance between the generated and true data distributions. Furthermore, this result confirms the intuition that confining the generative process to the dataset's intrinsic dimensionality is the most effective approach for low-rank data. This strategy is not only computationally more efficient than running the diffusion in the ambient space, but also enhances generation quality by avoiding the noise introduced by superfluous dimensions. The assumption of exact low-rank data is sometimes relaxed in the literature to an assumption on a decaying spectrum of eigenvalues. In the setting presented in this section where we plug-in the estimated covariance matrix into the true score, running the backward diffusion in the full data dimension is always optimal when the data has decaying but nonzero eigenvalues. In Section 5, we move to a setting closer to practice, where the score is learned by minimizing a constrained empirical risk. We will then prove the existence of a smaller optimal dimension for data with a decaying spectrum.

### 4.3. Generalization to arbitrary Gaussian distributions

We now explain how to generalize some of our preceding analysis from Gaussian distributions with diagonal covariance matrices to the more general case $p_0 = \mathcal{N}(0, \Sigma)$ for arbitrary $\Sigma$, and the backward processes with the new general data distribution $p_0$. Our goal is to establish a result

analogous to Proposition 4.2, that is, to characterize the optimal latent dimension given a stopping time of the diffusion process. To this end, let $\Sigma = O\Lambda O^\top$ be the eigen decomposition of $\Sigma$, where $O$ is an orthogonal matrix and $\Lambda$ is the diagonal matrix of eigenvalues, which we assume are distinct and ordered $\sigma_1^2 > \ldots > \sigma_D^2 > 0$. As in Section 4, we define a time partition by setting $t_1 = 0$ and $t_{D+1} = T$, and defining the intermediate timesteps for $d \in \{2, \ldots, D\}$ as:

$$t_d = T - \bar{a}^{-2}\left(\frac{3\sigma_d^2}{(1-\sigma_d^2)_+}\right),$$

where $\bar{a}^{-2}$ is given in (7). This definition, combined with the ordering of the eigenvalues, yields a sequence $0 = t_1 \leq t_2 \leq \cdots \leq t_D \leq t_{D+1} = T$. We show next that for this general Gaussian case, PCA projection onto $d$ components is optimal precisely within the interval $[t_d, t_{d+1})$.

**Proposition 4.4.** *For $2 \leq d \leq D$ and $t \in [t_d, t_{d+1})$, we have*

$$d = \underset{d' \in \{1,\ldots,D\}}{\arg\min} \Delta_{d,t,O}(\Sigma).$$

However, in practical applications, one rarely has access to the true underlying covariance matrix $\Sigma$ or its eigenbasis $O$. Instead, one must rely on estimations derived from observed data, where PCA is commonly used. Denote $\hat{\Sigma} = \frac{1}{n}\sum_{i=1}^n X_i X_i^\top$ to be the empirical covariance matrix. Applying a spectral decomposition yields $\hat{\Sigma} = \hat{O}\hat{\Lambda}\hat{O}^\top$, where $\hat{O}$ contains the orthonormal eigenvectors and $\hat{\Lambda} = \text{diag}(\hat{\sigma}_1^2, \ldots, \hat{\sigma}_D^2)$ where $\hat{\sigma}_D^2 < \ldots < \hat{\sigma}_1^2$ are the corresponding eigenvalues. Denote $S(\Sigma) = \sum_{d'=1}^D \max(\sigma_d, \sigma_d^2)$. For $u \geq 0$ and $d \in \{2, \ldots, D\}$, we let $\hat{T}_d(u) = T - \tau_d^-(u)$ and $\hat{t}_d(u) = T - \tau_d^+(u)$ where

$$\tau_d^\pm(u) = \bar{a}^{-2}\left(\frac{\hat{\sigma}_d^2 \pm 4S(\Sigma)\varepsilon_u + 2\hat{\sigma}_d\sqrt{\hat{\sigma}_d^2 - 4S(\Sigma)\varepsilon_u}}{(1-\hat{\sigma}_d^2)_+}\right),$$

where $\varepsilon_u = \frac{8C}{3}\left(\sqrt{\frac{D+u}{n}} + \frac{D+u}{n}\right)$. We assume that $\varepsilon_u$ is sufficiently small (i.e., $n$ large enough) so that the square root in the definition above is well-defined and the argument of $\bar{a}^{-2}$ is positive. By convention, we set $\hat{T}_1(u) = 0$ and $\hat{t}_{D+1}(u) = T$. Thus, for small $\varepsilon_u$, these timesteps are ordered as

$$0 = \hat{T}_1(u) < \hat{t}_2(u) < \hat{T}_2(u) < \cdots < \hat{t}_{D+1}(u) = T.$$

We are now in a position to describe the optimal projection strategy at each stopping time.

**Proposition 4.5.** *For $d \in \{1, \ldots, D\}$ and any $t \in [\hat{T}_d(u), \hat{t}_{d+1}(u)]$, with probability $1 - 2e^{-u}$,*

$$d = \underset{d' \in \{1,\ldots,D\}}{\arg\min} \Delta_{d,t,\hat{O}}(\hat{\Sigma}).$$

This proposition generalizes the result of Proposition 4.2 to the case of a general Gaussian data distribution. The analysis reveals that, for any latent dimension $d$, there exists a time interval where a $d$-dimensional projected diffusion process minimizes the distance to the target distribution with high probability. Notably, this result is consistent with our previous conclusions. In the idealized scenario where the variance estimation error is zero (i.e., $\varepsilon_u = 0$, implying $\hat{\Sigma} = \Sigma$,) the formula for the optimal time $\hat{t}_d = \hat{T}_d$ simplifies precisely to the one derived in Proposition 4.4.

## 5. Performance of the score matching ERM

In the previous section, we analyzed the properties of diffusion processes with a score tailored to independent Gaussian distributions involving either exact or plugged-in estimated variances. In practice, the score is rather *learned* by solving a regression problem called score matching. Specifically, given a training sample $(X_1, \ldots, X_n)$ independently drawn from the data distribution $p_0$, the empirical score matching objective writes

$$\mathcal{R}(s) = \frac{1}{n}\sum_{i=1}^n \mathbb{E}_{t\sim\mathcal{T},\varepsilon\sim\mathcal{N}(0,I_D)}\left\|s(b_t X_i + a_t\varepsilon, t) + \frac{\varepsilon}{a_t}\right\|^2 \tag{8}$$

for some absolutely continuous distribution $\mathcal{T}$ with positive mass over $[0, T]$, and where the predictor $s : \mathbb{R}^D \times \mathbb{R} \to \mathbb{R}$ belongs to some hypothesis class $\mathcal{F}_C$, typically a neural network architecture. In our context, recall that the score function of a Gaussian distribution with diagonal covariance $\Sigma$ is

$$\nabla \log p_t(x) = -(a_t^2 I_D + b_t^2\Sigma)^{-1}x, \tag{9}$$

which takes the form of a time-dependent diagonal matrix multiplied by $x$. Thus a natural choice of hypothesis class given the form of the true score function (9) is, for $C > 1$,

$$\mathcal{F}_C = \big\{s_M : \mathbb{R}^d \times \mathbb{R}_+ \to \mathbb{R}^d : s_M(x,t) = -M(t)x,$$
$$M(t) = \text{diag}(m_1(t), \ldots, m_D(t))$$
$$m_i \in \mathcal{L}_2(\mathbb{R}_+, \mathbb{R}), \|m_i\|_\infty < C\big\}.$$

The assumption of $C > 1$ is essential since we start the backward diffusion from a standard Gaussian distribution whose score function is the identity function. We introduce the norm constraint on the weights to account for two phenomena. First, the norm of the true score function (9) blows up for times close to 0 (in particular if the covariance matrix is singular or close to singular), which is known to create numerical instabilities (Lu et al., 2023; Yang et al., 2023). This is mitigated in practice for instance by early stopping the diffusion. Here, we implement this mitigation by capping the weight norm. Second, it is known that gradient descent has an implicit bias towards learning low-norm solutions. Although quantifying this effect is beyond the scope of this paper, the explicit weight constraint provides an analytically

tractable analogue. More precisely, one can easily derive the following explicit formula for the minimizer of the score matching over $\mathcal{F}_C$.

**Proposition 5.1.** *Let $\hat{\sigma}_d^2 = \frac{1}{n}\sum_{i=1}^n X_{id}^2$ be the empirical variance for the $d$-th component of the training data. Then the minimizer of the score matching objective* (8) *over $\mathcal{F}_C$ is given by $\hat{M}(t) = \mathrm{diag}(\hat{m}_1(t), \ldots, \hat{m}_D(t))$ where, for $d \in \{1, \ldots, D\}$,*

$$\hat{m}_d(t) = \min\left(C, \frac{1}{a_t^2 + b_t^2 \hat{\sigma}_d^2}\right).$$

Our goal in the following is to characterize the optimal latent dimension when using the score defined by Proposition 5.1. For this purpose, as before, we quantify the distance between the data distribution and the distribution generated by the backward process for $d \in \{1, \ldots, D\}$. While we do not consider early stopping here, to focus on the influence of the regularization parameter $C$ on the choice of the latent dimensionality, we note that equation (12) in Appendix A gives the formulas for the distribution of the backward diffusion at any time $t > 0$, from which the Fréchet distance for the early-stopped process could be derived and numerically studied. For simplicity, we keep the data distribution $p_0$ to be a Gaussian distribution with independent component, and specialize to the Ornstein-Uhlenbeck process. In this case, the sample $\overleftarrow{X}_t$ are generated by the backward SDE for $t \in [0, T]$

$$d\overleftarrow{X}_t = (\overleftarrow{X}_t + 2s_{\hat{M}}(\overleftarrow{X}_t, T-t))dt + \sqrt{2}d\overleftarrow{W}_t,$$
$$\overleftarrow{X}_0 \sim \mathcal{N}(0, I_D).$$

Note that we consider the standard setting in which the backward process starts from a standard Gaussian. We can then characterize the optimal projection for the latent diffusion, as shown next.

**Proposition 5.2.** *Define $1 \le d_1 \le d_2 \le D$ as follows:*

$$d_1 = \max\{d' \in \{1, \ldots, D\} : 1/C \le \hat{\sigma}_{d'}^2\}$$

*and*

$$d_2 = \min\left\{d' \in \{1, \ldots, D\} : \frac{1}{2C-1} \ge 4\sigma_{d'}^2\right\}.$$

*(If the corresponding set in their definition is empty, we let $d_1 = 1$ and $d_2 = D$, respectively.) Then, with high probability, there exists an optimal projection dimension $d_1 \le d_{\min} \le d_2$ such that*

$$d_{\min} = \underset{d' \in \{1, \ldots, D\}}{\arg\min} \; d_F(P_{d'}^\top P_{d'} \overleftarrow{X}_T, \overrightarrow{X_0}).$$

To gain intuition into Proposition 5.2, let us consider some illustrative cases depending on the weight constraint $C$ (for

the full derivation of these cases, see Appendix B.8). First, when $C = \infty$ and the data covariance is non-singular, we get $d_1 = d_2 = D$, thus Proposition 5.2 suggests to take the projection matrix $I_D$, which is expected since $\mathcal{F}_C$ is then large enough to contain the true empirical score function. Second, consider the scenario when the data distribution lies on a linear subspace of dimension $d_0$. If $C$ is large enough, we obtain $d_1 = d_2 = d_0$, meaning that the projection onto the data subspace is the optimal sampling strategy, which is in line with Proposition 4.3. Finally, the optimal projection can also be made explicit for exponentially-decaying covariance spectrum.

**Corollary 5.3.** *Let $\lambda > 16$. Assume that $\Sigma = \mathrm{diag}(\lambda^{-1}, \ldots, \lambda^{-D})$ and $\lambda \le C \le \lambda^D$. Let $d \in \{1, \ldots, D\}$ be such that $\hat{\sigma}_{d+1}^2 \le 1/C \le \hat{\sigma}_d^2$. Then, with $n$ large enough and high probability,*

$$d_{\min} \in \{d, d+1\}.$$

Interestingly, when the covariance structure decays exponentially, the capacity of the score-function class—captured here by the parameter $C$—is directly tied to the optimal projection dimension. The latter scales logarithmically in the parameter $C$ since $\hat{\sigma}_{d_{\min}}^2 = \lambda^{-d_{\min}} \approx 1/C$. An analogous result can be derived for covariance structures with power-law decay. We verify this prediction empirically: we generate data from a 10-dimensional Gaussian with eigenvalues $\Sigma = \mathrm{diag}(\lambda^{-1}, \ldots, \lambda^{-10})$, train a linear score model with clipped parameters in $\mathcal{F}_C$, and run the backward diffusion for different projection dimensions. As shown in Figure 4, the optimal projection dimension matches the prediction of Corollary 5.3.

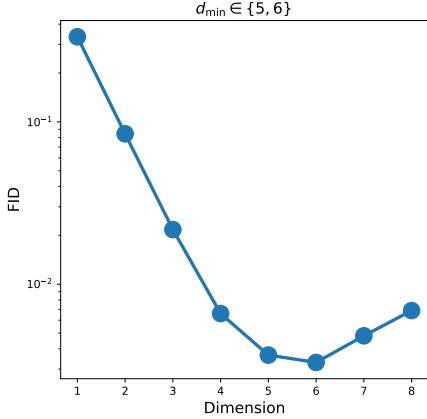

*Figure 4.* Fréchet distance for a 10-dimensional Gaussian with exponentially decaying eigenvalues and clipped linear score.

# 6. Empirical analysis

We now evaluate our theoretical predictions through experiments of increasing complexity, moving from synthetic data

to real images with non-linear architectures.

## 6.1. GMM with exact scores

To isolate the phenomena predicted by our theory from potential confounds due to score estimation error or Gaussianity of the data, we conduct an ablation study on Gaussian Mixture Model (GMM) data using the exact score function. We employ very small discretization timesteps to avoid numerical instability, thereby examining the diffusion trajectory under near-ideal conditions. As shown in Figure 5, even in this setting, the FID trajectories exhibit the characteristic U-shape, and the optimal latent dimensionality remains dependent on the stopping time. This confirms that these phenomena are fundamental to the geometry of the latent space and not artifacts of score estimation or the Gaussian assumption.

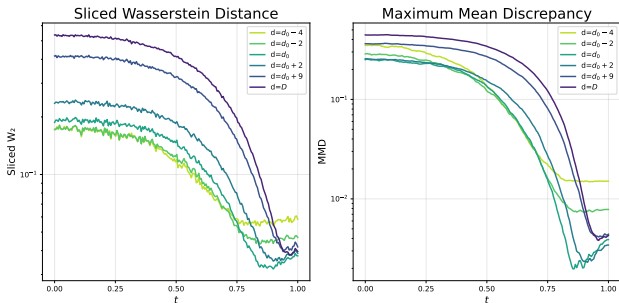

*Figure 5.* Ablation on GMM data with exact score and fine discretization.

## 6.2. Linear autoencoders on MNIST

To bridge the gap between our Gaussian theory and practical settings with real data, we train linear autoencoders on MNIST (Deng, 2012) with latent dimensions 64, 256, and 400, followed by corresponding U-Net LDMs. As shown in Figure 6, the results are consistent with Proposition 4.2: for each LDM, there exists a time interval in which it achieves the lowest FID, and at later diffusion times a larger latent dimension is required. This confirms that the dimension–stopping time trade-off extends beyond the Gaussian setting to real image distributions, when using linear encoders.

## 6.3. Noisy AE proxy on ImageNet

We now evaluate our framework on natural images with non-linear architectures, moving beyond the Gaussian and linear settings of the previous sections. Our theoretical analysis (Section 3) suggests that the backward diffusion should behave similarly to a *noisy AE*: for a sample $x \in \mathbb{R}^D$ with encoder $E : \mathbb{R}^D \to \mathbb{R}^d$ and decoder $D : \mathbb{R}^d \to \mathbb{R}^D$, the noisy AE computes $D(b_t E(x) + a_t Z)$ where $Z \sim \mathcal{N}(0, I_d)$, simulating the diffusion corruption at time $t$ within the latent space before decoding. In the Gaussian linear setting

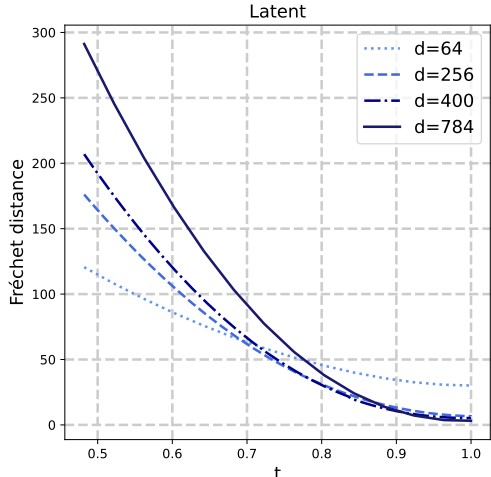

*Figure 6.* FID trajectories for linear autoencoders on MNIST with latent dimensions 64, 256, and 400, each paired with a U-Net LDM.

of Sections 4–5, this noisy AE is exactly equivalent to the LDM. We investigate whether this equivalence extends approximately to practical settings.

**Custom-trained AEs on ImageNet-256.** We train U-Net-based AEs with latent resolutions $32^2 \times 4$, $64^2 \times 3$, and $32^2 \times 16$ on ImageNet-256 (Deng et al., 2009), along with corresponding LDMs. To ensure a controlled comparison, we fix the combined parameter count (AE + LDM) at 500M across all configurations.

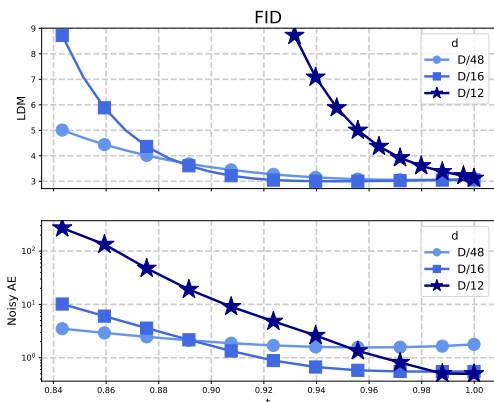

*Figure 7.* FID trajectories for LDMs and noisy AE proxies on ImageNet-256 with latent dimensions $32 \times 32 \times 4$, $64 \times 64 \times 3$, and $32 \times 32 \times 16$. Additional metrics are in Appendix E.

Figure 7 presents the FID trajectories for both LDMs and their noisy AE counterparts. We observe two key phenomena. First, the FID curves exhibit a strict alignment in performance rankings: the intersection points across different latent dimensions are identical for LDMs and noisy AEs, confirming that the proxy faithfully captures the relative or-

dering. Consistent with our theory, larger latent dimensions require later stopping times. Second, the optimal quality is not achieved at full denoising ($t = T$) but at an earlier time $t < T$ (Figure 8), likely due to high-frequency decoder artifacts (Odena et al., 2016). Crucially, this optimal stopping time for the LDM coincides with the time minimizing the noisy AE's FID. Similar findings for CelebA-HQ (Liu et al., 2015) are detailed in Appendix E.

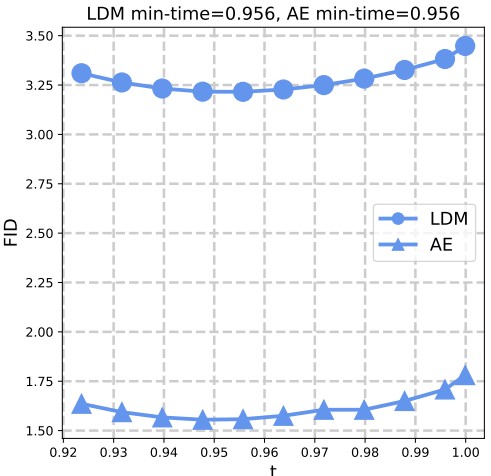

*Figure 8.* Zoom-in of Figure 7 for latent dimension $32{\times}32{\times}4$.

**Pre-trained Stable Diffusion VAE.** To demonstrate the practical utility of our framework beyond custom-trained models, we evaluate the noisy AE proxy using the off-the-shelf Stable Diffusion VAE (Podell et al., 2024) on ImageNet-256. As shown in Figure 9, the noisy AE proxy closely mimics the generative trajectory of the corresponding LDM, even with frozen, pre-trained representations.

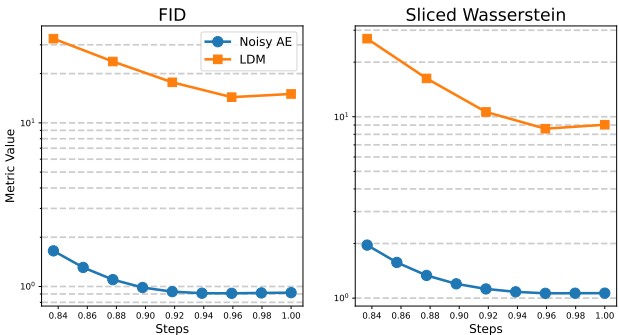

*Figure 9.* Noisy AE proxy with the pre-trained SD-VAE on ImageNet-256.

**FLUX autoencoder.** We further extend the evaluation to the FLUX-AE (Labs et al., 2025), a state-of-the-art autoencoder architecture. As shown in Figure 10, the noisy AE proxy continues to show the U-shape behavior.

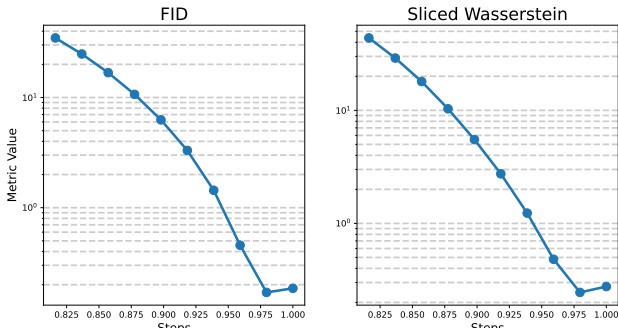

*Figure 10.* Noisy AE proxy with the pre-trained FLUX-AE on ImageNet-256.

**Summary.** Overall, our experiments confirm key theoretical predictions: (i) the optimal latent dimensionality depends on the diffusion stopping time, (ii) FID curves exhibit a U-shape indicating the benefit of early stopping, and (iii) the noisy AE proxy closely tracks LDM performance. The strong alignment between LDMs and noisy AEs suggests that the latter can serve as an efficient proxy for model selection, potentially allowing one to identify optimal hyperparameters—such as stopping time and latent dimensionality—without the computational cost of training full LDMs.

While we focus on FID as the primary metric, we additionally measure generation quality with MMD and sliced Wasserstein distance (MIND, Berthet et al., 2026) (Appendix E), observing consistent U-shaped behavior. Visual inspection further confirms that LDM-generated samples evolve only marginally during the final diffusion steps, whereas pixel-space models undergo pronounced refinement in the same interval.

## 7. Conclusion

This paper provides a theoretical analysis of optimal stopping time in latent diffusion models, showing its critical dependence on latent space dimensionality and its interaction with other hyperparameters of the diffusion process, such as weight regularization in the score matching phase. Our results focus on Gaussian distributions, given their tractability and prominence in prior theoretical works (Pierret & Galerne, 2024; Hurault et al., 2025). Taken together, these insights open compelling research directions, for deepening the theoretical properties of latent diffusion models and assessing when they can match or surpass the sampling quality of standard diffusion models.

## Acknowledgments

Authors thank Samuel Hurault for fruitful discussions. P.M. is supported by a Google PhD Fellowship.

## Impact Statement

This paper presents fundamental research in machine learning. There are many potential societal consequences of our work, none of which we feel must be specifically highlighted here.

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

# Appendix

## A. Discussion on diffusion and autoencoders

Assume that the data distribution $p_0$ is centered Gaussian $\mathcal{N}(0, \Sigma)$. Then $P\overleftarrow{X}_{T-t}$ and $P\overrightarrow{X}_t$ also follow Gaussian distributions

$$P\overleftarrow{X}_{T-t} \overset{\mathcal{D}}{=} P\overrightarrow{X}_t \sim \mathcal{N}(0, P_d(a_t^2 I_D + b_t^2 \Sigma)P_d^\top) = \mathcal{N}(0, a_t^2 I_d + b_t^2 P_d \Sigma P_d^\top), \tag{10}$$

where the covariance matrix of $P_d\overleftarrow{X}_{T-t}$ effectively zeros out the last $D - d$ dimensions and $a_t = \sqrt{1 - b_t^2}$ while $b_t = e^{-\int_0^t w_t^2 dt}$. A typical choice is the Ornstein-Uhlenbeck process, where $w_t \equiv 1$, which implies $a_t = \sqrt{1 - e^{-2t}}$ and $b_t = e^{-t}$. When $p_0$ is a Gaussian distribution, the score function $\nabla \log p_t$ is completely determined by the covariance matrix $\Sigma$. Indeed, in (10) we have

$$s_{P_d}(x, t) = -(a_t^2 I_d + b_t^2 P_d \Sigma P_d^\top)^{-1} x, \quad x \in \mathbb{R}^d.$$

Therefore, we may simplify the learning of the score function to the task of covariance matrix estimation. Since we assumed that the components of $p_0$ are independent, we consider a class of estimators consisting of diagonal matrices $\hat{\Sigma} = \text{diag}(\hat{\sigma}_1^2, \ldots, \hat{\sigma}_D^2)$, where its diagonal elements are the estimated variances given by $\frac{1}{n} \sum_{i=1}^n X_{id}^2$ for all $d \in \{1, \ldots, D\}$ where $X_i = (X_{i1}, \ldots, X_{iD}) \in \mathbb{R}^D$. Furthermore, we assume $\hat{\sigma}_1 \geq \ldots \geq \hat{\sigma}_D > 0$, which is satisfied with high probability when $n$ is large enough. Plugging the estimated covariance matrix in the final distribution of $P_d\overrightarrow{X}_T$ and in the score function $s_{P_d}$, we can define the estimated sampling procedure

$$dP\overleftarrow{\hat{X}}_t = (w_{T-t}^2 P\overleftarrow{\hat{X}}_t + 2w_{T-t}^2 \hat{s}_P(P\overleftarrow{\hat{X}}_t, T - t))dt + \sqrt{2w_{T-t}^2} dP\overleftarrow{W}_t, \quad P\overleftarrow{\hat{X}}_0 \sim (P)_\# \hat{p}_T, \tag{11}$$

where

$$\hat{s}_P(x, t) = -(a_t^2 I_d + b_t^2 P\hat{\Sigma}P^\top)^{-1} x \quad \text{and} \quad (P)_\# \hat{p}_T \sim \mathcal{N}(0, a_T^2 I_d + b_T^2 P\hat{\Sigma}P^\top).$$

By the same derivations as above, we have the following identity

$$P\overleftarrow{\hat{X}}_t \sim \mathcal{N}(0, a_{T-t}^2 I_d + b_{T-t}^2 P\hat{\Sigma}P^\top). \tag{12}$$

Therefore, our primary quantity of interest $\Delta_{d,t,O}(\Sigma)$ (resp. $\Delta_{d,t}(\hat{\Sigma})$) corresponds to the distance of the distribution of $OP_d^\top P_d O^\top \overleftarrow{X}_t$ (resp. $OP_d^\top P_d O^\top \overleftarrow{\hat{X}}_t$) to the true distribution $p_0$.

## B. Proofs of results

### B.1. Proof of Proposition 4.1

We show the equivalent statement: $t \in [0, T] \mapsto \Delta_{d,t}(\hat{\Sigma})$ is non-decreasing if and only if (6) holds. We start by calculating the Fréchet distance $\Delta_{d,t}(\hat{\Sigma})$ by using (5):

$$\Delta_{d,t}(\hat{\Sigma}) = \sum_{d'=d+1}^D \sigma_{d'}^2 + \sum_{d'=1}^d \left( b_t^2 \hat{\sigma}_{d'}^2 + a_t^2 + \sigma_{d'}^2 - 2\sigma_{d'}\sqrt{a_t^2 + b_t^2 \hat{\sigma}_{d'}^2} \right)$$

$$= \sum_{d'=d+1}^D \sigma_{d'}^2 + \sum_{d'=1}^d \left( \sqrt{a_t^2 + (1 - a_t^2)\hat{\sigma}_{d'}^2} - \sigma_{d'} \right)^2.$$

Since $t \mapsto a_t^2$ is strictly increasing with $a_0 = 0$, the monotonicity of $\Delta_{d,t}(\hat{\Sigma})$ with respect to $t$ is equivalent to the monotonicity with respect to $a_t^2$. By considering the function $f : [0, a_T^2] \to \mathbb{R}$ defined by

$$f(x) = \sum_{d'=1}^d \left( \sqrt{x + (1 - x)\hat{\sigma}_{d'}^2} - \sigma_{d'} \right)^2,$$

we see that $\Delta_{d,t}(\hat{\Sigma})$ is non-decreasing if and only if $f$ is non-decreasing. Additionally,

$$f'(x) = \sum_{d'=1}^d \left( 1 - \frac{\sigma_{d'}}{\sqrt{x + (1 - x)\hat{\sigma}_{d'}^2}} \right)(1 - \hat{\sigma}_{d'}^2),$$

and

$$f''(x) = \sum_{d'=1}^{d} \frac{\sigma_{d'}(1 - \hat{\sigma}_{d'}^2)^2}{2(x + (1-x)\hat{\sigma}_{d'}^2)^{3/2}} > 0.$$

Hence, $f$ is convex so it is non-decreasing if and only if $f'(0) \geq 0$. Therefore,

$$\Delta_{d,t}(\hat{\Sigma}) = f(a_t^2) + \sum_{d'=d+1}^{D} \sigma_{d'}^2$$

is non-decreasing if and only if $f'(0) \geq 0$, i.e., if and only if $\sum_{d'=1}^{d}(1 - \frac{\sigma_{d'}}{\hat{\sigma}_{d'}})(1 - \hat{\sigma}_{d'}^2) \geq 0$. This shows the second statement of the proposition. The monotonicity of $\Delta_{d,t}(\Sigma)$ can be shown by replacing $\hat{\sigma}_{d'}$ with $\sigma_{d'}$ in the derivative $f'$, which is 0 when $a_t = 0$.

## B.2. Proof of Proposition 4.2

The first part of Proposition 4.2 concerns the minimization of $\Delta_{d,t}(\Sigma)$. Recall that $t_d = T - \bar{a}^{-2}\left(\frac{3\sigma_d^2}{1-\sigma_d^2}\right)$. To prove that $\Delta_{d,t}(\Sigma)$ achieves the minimum for $t \in [t_d, t_{d+1})$ (where the time interval is fixed), we will demonstrate how the distance $\Delta_{d,t}(\Sigma)$ behaves as a function of the projection dimension $d$. Specifically, we aim to show that, for any $d \in \{2, \ldots, D\}$,

$$\Delta_{d,t}(\Sigma) \leq \Delta_{d-1,t}(\Sigma) \quad \text{iff } t \geq t_d. \tag{13}$$

This inequality in turn implies that for a given $t$ in a fixed interval $[t_d, t_{d+1})$, the minimum distance $\Delta_{d,t}(\Sigma) = d_F^2(P_d^\top P_d \overrightarrow{X}_t, \overrightarrow{X}_0)$ is attained by the projected process $P_d \overrightarrow{X}_t$ in dimension $d$. To establish them, we first explicitly compute the Fréchet distance $\Delta_{d,t}(\Sigma)$. Recall that the Fréchet distance between two zero-mean Gaussian distributions $\mathcal{N}(0, \Sigma_1)$ and $\mathcal{N}(0, \Sigma_2)$ is given by $\text{Tr}(\Sigma_1 + \Sigma_2 - 2(\Sigma_1^{1/2}\Sigma_2\Sigma_1^{1/2})^{1/2})$, and that the covariance matrix of $P_d \overleftarrow{X}_t$ is equal to $P_d(a_{T-t}^2 I_d + b_{T-t}^2 \Sigma)P_d$. Therefore, it is possible to calculate the Fréchet distance to the target for the projected processes directly, as, for any $d \in \{1, \ldots, D\}$,

$$\Delta_{d,t}(\Sigma) = \sum_{j=1}^{D} \sigma_j^2 + \sum_{j=1}^{d}(a_{T-t}^2 + b_{T-t}^2 \sigma_j^2) - 2\sum_{j=1}^{d} \sigma_j \sqrt{a_{T-t}^2 + b_{T-t}^2 \sigma_j^2},$$

so that

$$\begin{aligned}
\delta_{d,t} &:= d_F^2(P_d^\top P_d \overleftarrow{X}_t, \overrightarrow{X}_0) - d_F^2(P_{d-1}^\top P_{d-1} \overleftarrow{X}_t, \overrightarrow{X}_0) \\
&= \sqrt{\sigma_d^2 + a_{T-t}^2(1 - \sigma_d^2)}\left(\sqrt{a_{T-t}^2(1 - \sigma_d^2) + \sigma_d^2} - 2\sigma_d\right).
\end{aligned}$$

We see that $\delta_{d,t}$ has the same sign as the term in the parenthesis, which itself has the same sign as $a_{T-t}^2(1 - \sigma_d^2) - 3\sigma_d^2$. Then,

- if $\sigma_d \geq 1$ or $\frac{3\sigma_d^2}{1-\sigma_d^2} \geq a_T^2$, $\delta_{d,t}$ is non-positive for all $t \in [0, T]$, while $t_d = 0$ by definition;

- otherwise, $\delta_{d,t}$ is non-positive if and only if $a_{T-t}^2 \leq \frac{3\sigma_d^2}{1-\sigma_d^2}$ which is equivalent to

$$T - t \leq a^{-2}\left(\frac{3\sigma_d^2}{1 - \sigma_d^2}\right) = T - t_d.$$

Putting things together, we obtain that $\Delta_{d,t}(\Sigma) - d_F^2(P_{d-1}^\top P_{d-1} \overleftarrow{X}_t, \overrightarrow{X}_0)$ is non-positive iff $t \geq t_d$, which is exactly (13). The proof in the case of estimated variances can be derived in a similar fashion as long as the estimated variances $\hat{\sigma}_i$ and times $\hat{t}_i$ are well-ordered, which happens with high probability for a sufficiently large sample.

## B.3. Proof of Proposition 4.3

We first state the full proposition.

**Proposition B.1.** *Assume that* $\Sigma = \operatorname{diag}(\sigma^2, \ldots, \sigma^2, 0, \ldots, 0)$ *with the last* $D - d_0$ *entries equal to* $0$, *and the estimated variances are ordered as* $\hat{\sigma}_1^2 \geq \hat{\sigma}_2^2 \geq \ldots \geq \hat{\sigma}_{d_0}^2$. *Let* $\varepsilon \in (0, 1)$. *For*

$$t \in \left[ T - \bar{a}^{-2} \left( \frac{3 - \varepsilon}{1 + \varepsilon} \frac{\hat{\sigma}_1^2}{1 - \hat{\sigma}_1^2} \right), T \right),$$

*with probability* $1 - 2d_0 e^{-\frac{\varepsilon^2 n}{8}}$, *we have*

$$d_0 = \operatorname*{arg\,min}_{d' \in \{1, \ldots, D\}} \Delta_{d,t}(\hat{\Sigma}).$$

*If, in addition,*

$$\sum_{d'=1}^{d_0} \left( 1 - \frac{\sigma}{\hat{\sigma}_{d'}} \right)(1 - \hat{\sigma}_{d'}^2) < 0, \tag{14}$$

*then*

$$\sum_{d'=1}^{d_0} \left( 1 - \frac{\sigma}{\sqrt{\hat{\sigma}_{d'}^2 + (1 - \hat{\sigma}_{d'}^2) a_t^2}} \right)(1 - \hat{\sigma}_{d'}^2) = 0,$$

*has a unique solution which we denote by* $\hat{\delta}_{d_0}$. *By convention, if the condition* (14) *is not satisfied, we set* $\hat{\delta}_{d_0} = 0$. *Then, with probability* $1 - 2d_0 e^{-\frac{\varepsilon^2 n}{8}}$,

$$(d_0, T - \hat{\delta}_{d_0}) = \operatorname*{arg\,min}_{\substack{t \in [0, T] \\ d' \in \{1, \ldots, D\}}} \Delta_{d',t}(\hat{\Sigma}).$$

Let $\varepsilon \in (0, 1)$. We first note that according to Proposition C.1, by the union bound, with probability $1 - 2d_0 e^{-\frac{\epsilon^2 n}{8}}$ we have $|\sigma^2 - \hat{\sigma}_d^2| \leq \varepsilon \sigma^2$ for all $d \in \{1, \ldots, d_0\}$. We work under this event in the remainder of the proof. In particular, for all $d \in \{1, \ldots, d_0\}$, $\sigma_d^2 = \sigma^2 \geq \hat{\sigma}_1^2 / (1 + \varepsilon)$. Thus, by separating cases depending on whether $4\sigma_d^2 \leq 1$, a short calculation gives that

$$\min \left( 1, \frac{\frac{4}{1+\varepsilon} \hat{\sigma}_1^2 - \hat{\sigma}_1^2}{1 - \hat{\sigma}_1^2} \right) \leq \min \left( 1, \frac{4\sigma_d^2 - \hat{\sigma}_1^2}{1 - \hat{\sigma}_1^2} \right) \leq \frac{4\sigma_d^2 - \hat{\sigma}_d^2}{1 - \hat{\sigma}_d^2}.$$

Hence, by the monotonic increase of $\bar{a}^{-2}$,

$$\hat{t}_d = T - \bar{a}^{-2} \left( \frac{4\sigma_d^2 - \hat{\sigma}_d^2}{1 - \hat{\sigma}_d^2} \right)$$

$$\geq T - \bar{a}^{-2} \left( \frac{3 - \varepsilon}{1 + \varepsilon} \frac{\hat{\sigma}_1^2}{1 - \hat{\sigma}_1^2} \right).$$

Thus, $t \geq T - \bar{a}^{-2} \left( \frac{3-\varepsilon}{1+\varepsilon} \frac{\hat{\sigma}_1^2}{1-\hat{\sigma}_1^2} \right)$ implies $t \geq \hat{t}_d$ for every $d \in \{1, \ldots, d_0\}$. On the other hand, $t < T = \hat{t}_d$ for all $d \in \{d_0 + 1, \ldots, D\}$ since $\sigma_d = \hat{\sigma}_d = 0$. From here we deduce the desired result applying Proposition 4.2. In this second part, we study under the event where $|\sigma^2 - \hat{\sigma}_d^2| \leq \sigma^2$ for every $d \in \{1, \ldots, d_0\}$, which holds with probability $1 - 2d_0 e^{-n/8}$ by Proposition C.1. To prove the desired result, we first show that the minimum of the distance $\Delta_{d_0,t}(\hat{\Sigma})$ is attained at $t = T - \hat{\delta}_{d_0}$, as per its definition. We consider two cases depending on whether condition (14) is satisfied. First, if condition (14) holds, the proof of Proposition 4.1 establishes that $a_{T-\hat{\delta}_{d_0}}^2$ is the unique zero of the derivative $\frac{d}{da_t^2} \Delta_{d_0,t}(\hat{\Sigma})$. This confirms that $T - \hat{\delta}_{d_0}$ is the unique minimizer of the distance. Conversely, if condition (14) is not satisfied, then $\hat{\delta}_{d_0} = 0$. In this scenario, the squared distance $\Delta_{d_0,t}(\hat{\Sigma})$ is a non-increasing function of $t$ and thus attains its minimum at the endpoint $t = T$. We remark by Proposition 4.2 that, since $\hat{t}_d = T$ for every $d \in \{d_0 + 1, \ldots, D\}$, for every $t \in [0, T]$,

$$\Delta_{d_0, T - \hat{\delta}_{d_0}}(\hat{\Sigma}) \leq \Delta_{d_0, t}(\hat{\Sigma}) \leq \Delta_{d, t}(\hat{\Sigma}).$$

Observe that $\hat{t}_1 = \max_{d \in \{1,\dots,d_0\}} \hat{t}_d$, which is in the same order of $\hat{\sigma}_d$. Then from the proof of Proposition 4.2 we deduce that, for $t \geq \hat{t}_1$ and $d \in \{1, \dots, d_0\}$, that

$$\Delta_{d_0, T - \hat{\delta}_{d_0}}(\hat{\Sigma}) \leq \Delta_{d_0, t}(\hat{\Sigma}) \leq \Delta_{d, t}(\hat{\Sigma}).$$

If $\hat{t}_1 = 0$, the proof is finished. Note that this is the case if $\sigma^2 \geq 1/4$, since $\frac{4\sigma^2 - \hat{\sigma}_1^2}{(1 - \hat{\sigma}_1^2)_+} \geq 1$. We study from now the case where $\hat{t}_1 > 0$ and $\sigma^2 \leq 1/4$, with $t \leq \hat{t}_1$ and $d \in \{1, \dots, d_0\}$. In the remainder of the proof, we show, for $d \in \{1, \dots, d_0\}$, that $\Delta_{d, t}(\hat{\Sigma})$ is non-increasing on $[0, \hat{t}_1]$. This is equivalent to proving that $\Delta_{d, T-t}(\hat{\Sigma})$ is non-decreasing on $[T - \hat{t}_1, T]$. Recall that, as in the proof as Proposition 4.1,

$$\Delta_{d, T-t}(\hat{\Sigma}) = \sum_{d'=d+1}^{d_0} \sigma^2 + \sum_{d'=1}^{d} \left( \sqrt{a_t^2 + (1 - a_t^2)\hat{\sigma}_{d'}^2} - \sigma \right)^2.$$

Consider $f_d$ given by

$$f_d(x) = \sum_{d'=1}^{d} \left( \sqrt{x + (1 - x)\hat{\sigma}_{d'}^2} - \sigma \right)^2.$$

What we want to show is equivalent to $f$ being non-decreasing on $[a_{T-\hat{t}_1}^2, a_T^2]$. Since $f_d$ is convex as proven in Proposition 4.1, it is sufficient to show that $f'$ is positive at $a_{T-\hat{t}_1}^2$. All in all, since the derivative of $f_d$ is

$$f_d'(x) = \sum_{d'=1}^{d} \left( 1 - \frac{\sigma}{\sqrt{\hat{\sigma}_{d'}^2 + (1 - \hat{\sigma}_{d'}^2)x}} \right)(1 - \hat{\sigma}_{d'}^2),$$

if we are able to show that for any $d' \leq d_0$,

$$\left( 1 - \frac{\sigma}{\sqrt{\hat{\sigma}_{d'}^2 + (1 - \hat{\sigma}_{d'}^2)a_{T-\hat{t}_1}^2}} \right)(1 - \hat{\sigma}_{d'}^2) \geq 0, \tag{15}$$

then

$$f_d'(a_{T-\hat{t}}^2) = \sum_{d'=1}^{d} \left( 1 - \frac{\sigma}{\sqrt{\hat{\sigma}_{d'}^2 + (1 - \hat{\sigma}_{d'}^2)a_{T-\hat{t}_1}^2}} \right)(1 - \hat{\sigma}_{d'}^2) \geq 0.$$

The result above is twofold. First, we get that $\Delta_{d, T-t}(\hat{\Sigma})$ is increasing on the interval of interest. This also interestingly shows that the minimum of the Frobenius distance $t \mapsto \Delta_{d_0, t}(\hat{\Sigma})$ is reached after $\hat{t}_1$. Since by definition the minium is reached at $T - \hat{\delta}_{d_0}$, we get that $T - \hat{\delta}_{d_0} \geq \hat{t}$. The only thing remaining is to show (15). Recall that $\hat{t}_1 = T - \bar{a}^{-2}\left(\frac{4\sigma^2 - \hat{\sigma}_1^2}{1 - \hat{\sigma}_1^2}\right)$, and that we assumed $\hat{t}_1 > 0$, which implies $\frac{4\sigma^2 - \hat{\sigma}_1^2}{1 - \hat{\sigma}_1^2} < a_T^2$. On the other hand, recall that we work under the event that $|\sigma^2 - \hat{\sigma}_1^2| \leq \sigma^2$. Hence, $\hat{\sigma}_1^2 \leq 2\sigma^2 < 1$ and $\frac{4\sigma^2 - \hat{\sigma}_1^2}{1 - \hat{\sigma}_1^2} > 0$. Therefore, by definition of $\hat{t}_1$, we have

$$a_{T-\hat{t}_1}^2 = \frac{4\sigma^2 - \hat{\sigma}_1^2}{1 - \hat{\sigma}_1^2}.$$

From here, we prove (15). We rewrite (15) as

$$1 - \frac{\sigma}{\sqrt{\hat{\sigma}_{d'}^2 + (1 - \hat{\sigma}_{d'}^2)a_{T-\hat{t}}^2}} \geq 0.$$

$$\Leftrightarrow \quad (1 - \hat{\sigma}_{d'}^2)\frac{1 - 4\sigma^2}{1 - \hat{\sigma}_1^2} \leq 1 - \sigma^2.$$

Therefore, since $\hat{\sigma}_{d'} < 1$, we deduce that (15) is equivalent to showing:

$$\frac{1 - 4\sigma^2}{1 - \hat{\sigma}_1^2} \leq \frac{1 - \sigma^2}{1 - \hat{\sigma}_{d'}^2}.$$

To show this, recall the bound $\hat{\sigma}_1^2 \leq 2\sigma^2 < 1$. Thus,

$$\frac{1 - 4\sigma^2}{1 - \hat{\sigma}_1^2} \leq \frac{1 - 4\sigma^2}{1 - 2\sigma^2} = 1 - \sigma^2 \frac{2}{1 - 2\sigma^2} \leq 1 - \sigma^2 \leq \frac{1 - \sigma^2}{1 - \hat{\sigma}_{d'}^2},$$

which derives the desired inequality and we conclude the proof.

### B.4. Proof of Proposition 4.4

The proof follows by observing that the covariance matrix of $OP_d^\top P_d O^\top \overleftarrow{X}_t$ is given by

$$\mathrm{cov}[OP_d^\top P_d O^\top \overleftarrow{X}_t] = O\mathrm{diag}(a_{T-t}^2 + b_{T-t}^2 \sigma_1^2, \ldots, a_{T-t}^2 + b_{T-t}^2 \sigma_d^2, 0, \ldots, 0)O^\top.$$

Therefore, we have the following explicit form of the Fréchet distance between $OP_d^\top P_d O^\top \overleftarrow{X}_t$ and $\overrightarrow{X}_0$:

$$d_F(OP_d^\top P_d O^\top \overleftarrow{X}_t, \overrightarrow{X}_0) = \sum_{j=1}^{D} \sigma_j^2 + \sum_{j=1}^{d} (a_{T-t}^2 + b_{T-t}^2 \sigma_j^2) - 2\sum_{j=1}^{d} \sigma_j \sqrt{a_{T-t}^2 + b_{T-t}^2 \sigma_j^2}.$$

The proof is concluded by using the same argument as in the proof of Proposition 4.2.

### B.5. Proof of Proposition 4.5

Recall that $\hat{\Lambda} = \mathrm{diag}(\hat{\sigma}_1^2, \ldots \hat{\sigma}_D^2)$ the matrix of eigenvalues of the estimated covariance matrix $\hat{\Sigma} = \frac{1}{n}\sum_{i=1}^{n} X_i X_i^\top$. We first remark that

$$\begin{aligned}
\mathrm{cov}[\hat{O}P_d^\top P_d \hat{O}^\top \overleftarrow{X}_t] &= \hat{O}\mathrm{diag}(a_{T-t}^2 + b_{T-t}^2 \hat{\sigma}_1^2, \ldots, a_{T-t}^2 + b_{T-t}^2 \hat{\sigma}_d^2, 0, \ldots, 0)\hat{O}^\top \\
&= \hat{O}(a_{T-t}^2 P_d^\top P_d + b_{T-t}^2 P_d^\top P_d \hat{\Lambda})\hat{O}^\top.
\end{aligned}$$

Denote the covariance matrix of $\hat{O}P_d^\top P_d \hat{O}^\top \overleftarrow{X}_t$ by $\hat{\Sigma}_d(t)$. Recall that the Fréchet distance between two centered Gaussian distributions is

$$d_F^2(\mathcal{N}(0, \Sigma_1), \mathcal{N}(0, \Sigma_2)) = \mathrm{tr}(\Sigma_1 + \Sigma_2 - 2(\Sigma_2^{1/2}\Sigma_1\Sigma_2^{1/2})^{1/2}).$$

In the case of interest for us, we get

$$d_F^2(\hat{O}P_d^\top P_d \hat{O}^\top \overleftarrow{X}_t, \overrightarrow{X}_t) = \sum_{d'=1}^{D} \sigma_{d'}^2 + \sum_{d'=1}^{d} (a_{T-t}^2 + b_{T-t}^2 \hat{\sigma}_{d'}^2) - 2\mathrm{tr}((\hat{\Sigma}_d^{1/2}(t)\Sigma\hat{\Sigma}_d^{1/2}(t))^{1/2}).$$

We now argue that $\mathrm{tr}((\hat{\Sigma}_d^{1/2}(t)\Sigma\hat{\Sigma}_d^{1/2}(t))^{1/2})$ is approximately $\sum_{d'=1}^{d} \hat{\sigma}_{d'}\sqrt{a_{T-t}^2 + b_{T-t}^2 \hat{\sigma}_{d'}^2}$. By Proposition C.2, with probability $1 - 2e^{-u}$, we have $\Sigma \preceq \frac{1}{1 - \varepsilon_u}\hat{\Sigma}$, where $\preceq$ denotes the Loewner order. Hence,

$$\hat{\Sigma}_d^{1/2}(t)\Sigma\hat{\Sigma}_d^{1/2}(t) \preceq \frac{1}{1 - \varepsilon_u}\hat{\Sigma}_d^{1/2}(t)\hat{\Sigma}\hat{\Sigma}_d^{1/2}(t),$$

by Lemma D.1 (i). Since square root is a matrix monotonic function (see Lemma D.1 (ii)), we derive that

$$\begin{aligned}
\mathrm{tr}((\hat{\Sigma}_d^{1/2}(t)\Sigma\hat{\Sigma}_d^{1/2}(t))^{1/2}) &\leq \sqrt{\frac{1}{1 - \varepsilon_u}}\mathrm{tr}((\hat{\Sigma}_d^{1/2}(t)\hat{\Sigma}\hat{\Sigma}_d^{1/2}(t))^{1/2}) \\
&\leq (1 + \varepsilon_u)\mathrm{tr}((\hat{\Sigma}_d^{1/2}(t)\hat{\Sigma}\hat{\Sigma}_d^{1/2}(t))^{1/2}),
\end{aligned}$$

where we use $\varepsilon_u \leq 1/2$ in the last inequality. Then, by the commutativity of $\hat{\Sigma}_d(t)$ and $\hat{\Sigma}$,

$$\mathrm{tr}((\hat{\Sigma}_d^{1/2}(t)\hat{\Sigma}\hat{\Sigma}_d^{1/2}(t))^{1/2}) = \sum_{d'=1}^{d} \hat{\sigma}_{d'}\sqrt{a_{T-t}^2 + b_{T-t}^2 \hat{\sigma}_{d'}^2}.$$

By combining the results, we obtain

$$\text{tr}((\hat{\Sigma}_d^{1/2}(t)\Sigma\hat{\Sigma}_d^{1/2}(t))^{1/2}) \le (1+\varepsilon_u)\sum_{d'=1}^{d}\hat{\sigma}_{d'}\sqrt{a_{T-t}^2 + b_{T-t}^2\hat{\sigma}_{d'}^2}.$$

We may use the same argument to derive a similar lower bound, and thus deduce that

$$\left|\text{tr}((\hat{\Sigma}_d^{1/2}(t)\Sigma\hat{\Sigma}_d^{1/2}(t))^{1/2}) - \sum_{d'=1}^{d}\hat{\sigma}_{d'}\sqrt{a_{T-t}^2 + b_{T-t}^2\hat{\sigma}_{d'}^2}\right| \le \varepsilon_u\sum_{d'=1}^{d}\hat{\sigma}_{d'}\sqrt{a_{T-t}^2 + b_{T-t}^2\hat{\sigma}_{d'}^2}.$$

Note that if $\hat{\sigma}_{d'} \ge 1$, then $\sqrt{a_{T-t}^2 + b_{T-t}^2\hat{\sigma}_{d'}^2} \le \hat{\sigma}_{d'}$. Hence $\hat{\sigma}_{d'}\sqrt{a_{T-t}^2 + b_{T-t}^2\hat{\sigma}_{d'}^2} \le \hat{\sigma}_{d'}^2$. On the other hand, if $\hat{\sigma}_d < 1$, then $\sqrt{a_{T-t}^2 + b_{T-t}^2\hat{\sigma}_{d'}^2} \le 1$ and $\hat{\sigma}_{d'}\sqrt{a_{T-t}^2 + b_{T-t}^2\hat{\sigma}_{d'}^2} \le \hat{\sigma}_{d'}$. Therefore, by recalling that $S(\Sigma) = \sum_{d'=1}^{D}\max(\hat{\sigma}_{d'}, \hat{\sigma}_{d'}^2)$, we deduce that

$$\left|\text{tr}((\hat{\Sigma}_d^{1/2}(t)\Sigma\hat{\Sigma}_d^{1/2}(t))^{1/2}) - \sum_{d'=1}^{d}\hat{\sigma}_{d'}\sqrt{a_{T-t}^2 + b_{T-t}^2\hat{\sigma}_{d'}^2}\right| \le S(\Sigma)\varepsilon_u.$$

The Fréchet distance $d_F(\hat{O}P_d^\top P_d\hat{O}^\top\overleftarrow{\hat{X}}_t, \overrightarrow{X}_t)$ may now be bounded by

$$\left|d_F^2(\hat{O}P_d^\top P_d\hat{O}^\top\overleftarrow{\hat{X}}_t, \overrightarrow{X}_t) - \left(\sum_{d'=1}^{D}\sigma_{d'}^2 + \sum_{d'=1}^{d}(a_{T-t}^2 + b_{T-t}^2\hat{\sigma}_{d'}^2) - 2\sum_{d'=1}^{d}\hat{\sigma}_{d'}\sqrt{a_{T-t}^2 + b_{T-t}^2\hat{\sigma}_{d'}^2}\right)\right|$$
$$\le 2S(\Sigma)\varepsilon_u.$$

Hence, for $d \in \{2, \ldots, D\}$,

$$\left|d_F^2(\hat{O}P_d^\top P_d\hat{O}^\top\overleftarrow{\hat{X}}_t, \overrightarrow{X}_t) - d_F^2(\hat{O}P_{d-1}^\top P_{d-1}\hat{O}^\top\overleftarrow{\hat{X}}_t, \overrightarrow{X}_t)\right.$$
$$\left. - \sqrt{a_{T-t}^2 + b_{T-t}^2\hat{\sigma}_d^2}\left(\sqrt{a_{T-t}^2 + b_{T-t}^2\hat{\sigma}_d^2} - 2\hat{\sigma}_d\right)\right| \le 4S(\Sigma)\varepsilon_u.$$

We show in the following that if $t \ge \hat{T}_d(u)$, then

$$d_F^2(\hat{O}P_d^\top P_d\hat{O}^\top\overleftarrow{\hat{X}}_t, \overrightarrow{X}_t) \le d_F^2(\hat{O}P_{d-1}^\top P_{d-1}\hat{O}^\top\overleftarrow{\hat{X}}_t, \overrightarrow{X}_t).$$

Observe that,

$$d_F^2(\hat{O}P_d^\top P_d\hat{O}^\top\overleftarrow{\hat{X}}_t, \overrightarrow{X}_t) \le d_F^2(\hat{O}P_{d-1}^\top P_{d-1}\hat{O}^\top\overleftarrow{\hat{X}}_t, \overrightarrow{X}_t)$$
$$+ (\sqrt{a_{T-t}^2(1-\hat{\sigma}_d^2) + \hat{\sigma}_d^2} - \hat{\sigma}_d)^2 - \hat{\sigma}_d^2 + 4S(\Sigma)\varepsilon_u. \tag{16}$$

Hence, for $t$ such that the last term (16) is non-positive, we have $d_F^2(\hat{O}P_d^\top P_d\hat{O}^\top\overleftarrow{\hat{X}}_t, \overrightarrow{X}_t) \le d_F^2(\hat{O}P_{d-1}^\top P_{d-1}\hat{O}^\top\overleftarrow{\hat{X}}_t, \overrightarrow{X}_t)$. We now show that this is true when $t \ge \hat{T}_d(u)$. To do so, we split our argument in two cases. We first consider the scenario where $\hat{\sigma}_d \ge 1$. In this case, by definition, $\hat{T}_d(u) = 0$ and therefore we prove the result holds for all $t \in [0, T]$. Observe that $\sqrt{a_{T-t}^2(1-\hat{\sigma}_d^2) + \hat{\sigma}_d^2} \in [1, \hat{\sigma}_d]$, therefore

$$(16) \le (1-\hat{\sigma}_d)^2 - \hat{\sigma}_d^2 + 4S(\Sigma)\varepsilon_u = 1 - 2\hat{\sigma}_d + 4S(\Sigma)\varepsilon_u \le 1 - 2\hat{\sigma}_d + \hat{\sigma}_d \le 0,$$

where the last inequality holds for sufficiently small $\varepsilon_u$. Now we consider the case where $\hat{\sigma}_d < 1$, and hence $\sqrt{a_{T-t}^2(1-\hat{\sigma}_d^2) + \hat{\sigma}_d^2} \ge \hat{\sigma}_d$. Therefore,

$$(16) \le 0 \Leftrightarrow a_{T-t}^2 \le \frac{\hat{\sigma}_d^2 - 4S(\Sigma)\varepsilon_u + 2\hat{\sigma}_d\sqrt{\hat{\sigma}_d^2 - 4S(\Sigma)\varepsilon_u}}{1 - \hat{\sigma}_d^2},$$

and we conclude by observing that the last inequality is equivalent to $t \ge \hat{T}_d(u)$. We derive with a similar argument that if $t \le \hat{t}_d(u)$ then

$$d_F^2(\hat{O}P_d^\top P_d\hat{O}^\top\overleftarrow{\hat{X}}_t, \overrightarrow{X}_t) \ge d_F^2(\hat{O}P_{d-1}^\top P_{d-1}\hat{O}^\top\overleftarrow{\hat{X}}_t, \overrightarrow{X}_t),$$

and we conclude the proof.

### B.6. Proof of Proposition 5.1

We begin by rewriting the expression of the score matching objective:

$$\mathcal{R}(s_M) = \frac{1}{n}\sum_{i=1}^{n}\sum_{d=1}^{D}\mathbb{E}_{t\sim\mathcal{T},\varepsilon_d}\left[\left(m_d(t)(b_t X_{id}+a_t\varepsilon_d)-\frac{\varepsilon_d}{a_t}\right)^2\right].$$

To find the optimal $M(t)$, we note that the objective and the constraint are separable across $[0,T]$ and across dimensions $d \in \{1,\ldots,D\}$. Hence it suffices to minimize

$$r(m_d(t)) := \frac{1}{n}\sum_{i=1}^{n}\mathbb{E}_{\varepsilon_d}\left[\left(m_d(t)(b_t X_{id}+a_t\varepsilon_d)-\frac{\varepsilon_d}{a_t}\right)^2\right]$$

separately over $m_d(t) \in [-C, C]$ for each $t \in [0, T]$ and $d$. The derivative is

$$r'(m) = m\left(a_t^2 + b_t^2\frac{1}{n}\sum_{i=1}^{n}X_{id}^2\right) - 1.$$

Therefore, the minimum of $r$ over $[-C, C]$ is attained at

$$\hat{m}_d(t) = \min\left(C, \frac{1}{a_t^2 + b_t^2\hat{\sigma}_d^2}\right),$$

which concludes the proof.

### B.7. Proof of Proposition 5.2

Since the Fréchet distance is determined by the variance for centered random variables, the first step of the proof is to deduce the variance of $P_d\overleftarrow{X}_0$ for all $d$. Denote for $1 \le d \le D$, the variance of $\overleftarrow{X}_{t,d}$ by $V_{t,dd}$. It is known (see, for instance, Särkkä & Solin, 2019, Section 5.5) that $V_{t,dd}$ follows the following ODE:

$$\frac{dV_{t,dd}}{dt} = 2(1-2\hat{m}_d(T-t))V_{t,dd} + 2, \quad V_{0,dd} = 1. \tag{17}$$

An important intermediate step in this proof is to show the following:

$$(\sqrt{V_{t,dd}}-\sigma_d)^2 \le \sigma_d^2 \quad, \text{if} \quad d \le d_1, \tag{18}$$

$$(\sqrt{V_{t,dd}}-\sigma_d)^2 \ge \sigma_d^2 \quad, \text{if} \quad d \ge d_2. \tag{19}$$

To do so, we first develop an explicit expression for $V_{t,dd}$:

$$V_{t,dd} = \exp\left(\int_0^t 2(1-2\hat{m}_d(T-\tau))d\tau\right) + 2\int_0^t \exp\left(\int_s^t 2(1-2\hat{m}_d(T-\tau))d\tau\right)ds. \tag{20}$$

If $C < \frac{1}{a_0^2+b_0^2\hat{\sigma}_d^2} = \frac{1}{\hat{\sigma}_d^2}$, let $t_d'$ be the unique solution in $[0,T]$ of the equation $C = \hat{m}_d(T-t_d') = \frac{1}{a_{T-t_d'}^2+b_{T-t_d'}^2\hat{\sigma}_d^2}$. Otherwise, we set $t_d' = T$, which is always the case for $d \le d_1$. Remark that if $\hat{\sigma}_d \ge 1$, then $\frac{1}{a_t^2+b_t^2\hat{\sigma}_d^2} \le 1 \le C$. Thus, for such dimension $d$, we always have $t_d' = T$ and $d \le d_1$. We derive an explicit expression for the term $V_{T,dd}$. Recall that $a_t = \sqrt{1-e^{-2t}}$ and $b_t = e^{-t}$. Also note that $\hat{m}_d$ is decreasing on $[0,T]$, more precisely it is equal to $C$ on $[0, T-t_d']$ and equal to $1/(a_t^2+b_t^2\hat{\sigma}_d^2)$ for $t \in [T-t_d', T]$.

$$\exp\left(\int_s^T 2(1-2\hat{m}_d(T-\tau))d\tau\right)$$

$$= e^{2(T-s)}e^{-4C(T-s\vee t_d')}e^{-4(s\vee t_d'-s)}\left(\frac{1-(1-\hat{\sigma}_d^2)e^{-2(T-s\vee t_d')}}{1-(1-\hat{\sigma}_d^2)e^{-2(T-s)}}\right)^2, \tag{21}$$

where, in the last line, we use the following:

$$\int \hat{m}_d(\tau)d\tau = \tau + \frac{1}{2}\log\left(1 - (1 - \hat{\sigma}_d^2)e^{-2\tau}\right).$$

By substituting $s = 0$, we see that the first term in (20) is equal to

$$\exp\left(\int_0^T 2(1 - 2\hat{m}_d(T - \tau))d\tau\right) = e^{-2T}e^{-4(C-1)(T-t_d')}\left(\frac{1 - (1 - \hat{\sigma}_d^2)e^{-2(T-t_d')}}{1 - (1 - \hat{\sigma}_d^2)e^{-2T}}\right)^2. \tag{22}$$

Next, we focus on deriving an explicit expression of the second term in (20):

$$2\int_0^T \exp\left(\int_s^T 2(1 - 2\hat{m}_d(T - \tau))d\tau\right)ds$$

$$= \frac{1}{2C - 1}(1 - e^{(2-4C)(T-t_d')})$$

$$+ \frac{(1 - (1 - \hat{\sigma}_d^2)e^{-2(T-t_d')})e^{-4(C-1)(T-t_d')}}{1 - (1 - \hat{\sigma}_d^2)e^{-2T}}(e^{-2(T-t_d')} - e^{-2T}). \tag{23}$$

Therefore, by summing up the two terms (22) and (23), we deduce that

$$V_{T,dd} = \frac{1}{2C - 1}(1 - e^{(2-4C)(T-t_d')}) + e^{(2-4C)(T-t_d')}\frac{1 - (1 - \hat{\sigma}_d^2)e^{-2(T-t_d')}}{1 - (1 - \hat{\sigma}_d^2)e^{-2T}}\frac{1 - 2(1 - \hat{\sigma}_d^2)e^{-2T} + (1 - \hat{\sigma}_d^2)e^{-2(T+t_d')}}{1 - (1 - \hat{\sigma}_d^2)e^{-2T}}.$$

We remark that, for $d \le d_1$ we have $t_d' = T$ and we may simplify the expression of $V_{T,dd}$ to

$$V_{T,dd} = \hat{\sigma}_d^2\frac{1 - 2(1 - \hat{\sigma}_d^2)e^{-2T} + (1 - \hat{\sigma}_d^2)e^{-4T}}{1 - 2(1 - \hat{\sigma}_d^2)e^{-2T} + (1 - \hat{\sigma}_d^2)^2 e^{-4T}}. \tag{24}$$

Before we prove (18) and (19), we categorize the behavior of $V_{t,dd}$ according to the value of $\hat{\sigma}_d$ in the following lemma (delayed to the end of this proof).

**Lemma B.2.** *For $d \in \{1, \ldots, D\}$. If $\hat{\sigma}_d \ge 1$, then $V_{t,dd} \ge 1$ for every $t \in [0, T]$. If $\hat{\sigma}_d \le 1$, then $V_{t,dd} \le 1$ for every $t \in [0, T]$.*

Let us deduce from (24), for $d \le d_1$, $(\sqrt{V_{T,dd}} - \sigma_d)^2 \le \sigma_d^2$. We work under the high probability event that $|\sigma_d^2 - \hat{\sigma}_d^2| \le \sigma_d^2$ for every $d \in \{1, \ldots, D\}$, we split the proof into three cases:

- If $\sigma_d > 1$ then $\hat{\sigma}_d \ge 1$, from (24), we see that $V_{T,dd} < \hat{\sigma}_d^2 \le 4\sigma_d^2$, with high probability. Thus, $(\sqrt{V_{T,dd}} - \sigma_d)^2 \le \max((0 - \sigma_d)^2, (2\sigma_d - \sigma_d)^2) \le \sigma_d^2$.

- If $\sigma_d \in [\frac{1}{2}, 1)$ which implies that $\hat{\sigma}_d < 1$, then $V_{T,dd} \le 1$, we then have $(\sqrt{V_{T,dd}} - \sigma_d)^2 \le \max((0 - \sigma_d)^2, (1 - \sigma_d)^2) \le \sigma_d^2$.

- If $\sigma_d = 1$, we again split cases depending on whether $\hat{\sigma}_d \ge 1$. We get the same bounds as in the two previous cases.

- Finally, if $\sigma_d \le \frac{1}{2}$, with high probability, $|\sigma_d^2 - \hat{\sigma}_d^2| \le \sigma_d^2$. Hence, $\hat{\sigma}_d^2 \le 2\sigma_d^2 \le \frac{1}{2}$. Observing that the fraction in (24) is bounded by $1/(1 - \hat{\sigma}_d^2)$, we deduce that

$$V_{T,dd} \le \frac{\hat{\sigma}_d^2}{1 - \hat{\sigma}_d^2} \le 2\hat{\sigma}_d^2 \le 4\sigma_d^2, \quad \forall d \le d_1,$$

which gives the desired bound.

Next, for $d \ge d_2$, remark by definition of $t_d'$ that $\frac{1}{1-(1-\hat{\sigma}_d^2)e^{-2(T-t_d')}} = C$. Also note that the definition of $d_2$ and the fact that $C > 1$ implies that $\hat{\sigma}_d^2 < 1$. Hence,

$$V_{T,dd} = \frac{1}{2C - 1} + e^{(2-4C)(T-t_d')}\left(\frac{(1 - 2(1 - \hat{\sigma}_d^2)e^{-2T} + (1 - \hat{\sigma}_d^2)e^{-2(T+t_d')})}{C(1 - (1 - \hat{\sigma}_d^2)e^{-2T})^2} - \frac{1}{2C - 1}\right)$$

$$\geq \frac{1}{2C-1}.$$

Therefore, for $d \geq d_2$ we deduce that

$$V_{T,dd} \geq \frac{1}{2C-1} \geq 4\sigma_d^2.$$

To summarize, we derived the following bounds

$$(\sqrt{V_{T,dd}} - \sigma_d)^2 \leq \sigma_d^2, \quad \forall d \leq d_1,$$

and

$$(\sqrt{V_{T,dd}} - \sigma_d)^2 \geq \sigma_d^2, \quad \forall d > d_2.$$

By definition of the Fréchet distance, we have

$$d_F^2(P_d^\top P_d \overleftarrow{\tilde{X}}_T, \overrightarrow{X_0}) = \sum_{j=1}^{d} (\sqrt{V_{T,jj}} - \sigma_j)^2 + \sum_{j=d+1}^{D} \sigma_j^2,$$

we deduce that, for any $d < d_1 \leq d_2 < d'$,

$$d_F(P_{d_1}^\top P_{d_1} \overleftarrow{\tilde{X}}_T, \overrightarrow{X_0}) \leq d_F(P_d^\top P_d \overleftarrow{\tilde{X}}_T, \overrightarrow{X_0}),$$

and

$$d_F(P_{d_2}^\top P_{d_2} \overleftarrow{\tilde{X}}_T, \overrightarrow{X_0}) \leq d_F(P_{d'}^\top P_{d'} \overleftarrow{\tilde{X}}_T, \overrightarrow{X_0}).$$

Therefore, the minimum of $d_F(P_d^\top P_d \overleftarrow{\tilde{X}}_T, \overrightarrow{X_0})$ must occur between $d_1$ and $d_2$.

**Proof of Lemma B.2.**   Recall that $V_{t,dd}$ satisfies the ODE (17)

$$\frac{dV_{t,dd}}{dt} = 2(1 - 2\hat{m}_d(T-t))V_{t,dd} + 2, \quad V_{0,dd} = 1.$$

Assume that $\hat{\sigma}_d > 1$ and by contradiction that $V_{t,dd} < 1$ for some $t \in [0,T]$. Let $t_0 = \inf\{t : V_{t,dd} < 1\}$, by continuity, we have $V_{t_0,dd} = 1$. Then we have

$$\left[\frac{dV_{t,dd}}{dt}\right]_{t=t_0} = \frac{4(\hat{\sigma}_d^2 - 1)e^{-2t}}{1 - e^{-2t} + e^{-2t}\hat{\sigma}_d^2}.$$

Hence we have $[\frac{dV_{t,dd}}{dt}]_{t=t_0} > 0$ which contradicts the definition of $t_0$. Hence $V_{t,dd} \geq 1$ for all $t \in [0,T]$. The case for $\hat{\sigma}_d < 1$ can be derived similarly.

### B.8. Derivation of special cases of Proposition 5.2

First, consider the scenario where the learning capacity is unconstrained, effectively setting $C = \infty$, while the data covariance matrix is nonsingular. In this case, the condition on $d_1$ becomes $0 \leq \hat{\sigma}_d^2$, which is trivially satisfied for all $d \in \{1, \ldots, D\}$, implying $d_1 = D$. The condition for $d_2$ becomes $0 > 4\sigma_d^2$, which holds for none of $d$, thus implying $d_2 = D$. Therefore, when $C = \infty$, Proposition 5.2 entails that $d_{\min} = D$. Second, consider the scenario addressed in Proposition 4.3 where the true data distribution lies within a $d_0$-dimensional linear subspace, i.e., $\sigma_{d_0+1} = \cdots = \sigma_D = 0$ and $\sigma_1 = \cdots = \sigma_{d_0} = \sigma$. Assume that $C$ is sufficiently large to ensure that $1/C \leq \min(\sigma^2, \min_{d' \in \{1,\ldots,d_0\}} \hat{\sigma}_{d'}^2)$. Therefore, for $d \leq d_0$, one has $\frac{1}{C} \leq \hat{\sigma}_d^2$ (which is not satisfied anymore for $d$ beyond $d_0$), leading to $d_1 = d_0$. On the other hand, for $d > d_0$ we have $\frac{1}{2C-1} \geq 0 = 4\sigma_d$. Hence $d_0 = d_1 \leq d_2 \leq d_0$, which implies $d_2 = d_0$. Thus, Proposition 5.2 predicts $d_{\min} = d_0$. This suggests that the projection onto the subspace in which the data distribution lies is the optimal sampling strategy.

### B.9. Proof of Corollary 5.3

By the definition of $d$, we have $d_1 = d$. It remains to prove that $d_2 \le d + 1$. With $n$ large enough and high probability, we have $\hat{\sigma}_{d+1} \ge \sigma_{d+1}^2/2$. Therefore,

$$\frac{1}{4(2C-1)} \ge \frac{1}{8C} \ge \frac{\hat{\sigma}_{d+1}^2}{8} \ge \frac{\sigma_{d+1}^2}{16} = \frac{\lambda^{-(d+1)}}{16} \ge \lambda^{-(d+2)},$$

where we use the fact that $\lambda \ge 16$. This shows that $d_2 < d + 2$. Hence $d_2 \le d + 1$.

## C. Bounds on Gaussian estimation

In this section, we give some bounds for the estimation error for Gaussian distributions.

**Proposition C.1.** *Let $(X_1, \ldots, X_n)$ be sample drawn independently from $\mathcal{N}(0, \sigma^2)$. Then, for $\varepsilon > 0$, we have*

$$\mathbb{P}\Big[\Big|\frac{1}{n}\sum_{i=1}^{n} X_i^2 - \sigma_d^2\Big| \le \varepsilon\sigma_d^2\Big] \ge 1 - 2\exp\Big(-\frac{\varepsilon^2 n}{4(\varepsilon+1)}\Big).$$

*Proof.* By Ghosh (2021), if $Z \sim \chi^2(p)$ and $u > 0$,

$$\mathbb{P}[|Z - p| \ge u] \le 2\exp\Big(-\frac{u^2}{4(p+u)}\Big).$$

The result then unfolds from standard manipulations after observing that $\frac{1}{\sigma^2}\sum_{i=1}^{n} X_i^2$ follows a $\chi^2(n)$. $\square$

**Proposition C.2.** *Let $\Sigma$ be a semi-definite positive $D \times D$ matrix, and assume the sample $(X_1, \ldots, X_n)$ is drawn independently from $\mathcal{N}(0, \Sigma)$. Then, there is a universal constant $C$ such that, with probability $1 - 2e^{-u}$, the empirical covariance matrix $\hat{\Sigma} = \frac{1}{n}\sum_{i=1}^{n} X_i X_i^\top$ satisfies:*

$$-\frac{8C}{3}\Big(\sqrt{\frac{D+u}{n}} + \frac{D+u}{n}\Big)\Sigma \ldots \hat{\Sigma} - \Sigma \ldots \frac{8C}{3}\Big(\sqrt{\frac{D+u}{n}} + \frac{D+u}{n}\Big)\Sigma, where \ldots denotes the Loewner order.$$

*Proof.* It is shown in Vershynin (2018, Theorem 4.6.1) that, with probability $1 - 2e^{-u}$,

$$\|\Sigma^{-1/2}\hat{\Sigma}\Sigma^{-1/2} - I_D\|_{op} \le K^2 C\Big(\sqrt{\frac{D+u}{n}} + \frac{D+u}{n}\Big),$$

where $\|\cdot\|_{op}$ denotes the operator norm and $K$ is a constant satisfying

$$\|X^\top x\|_{\psi_2} \le K\|X^\top x\|_{L_2}, \forall x \in \mathbb{R}^D,$$

where $\|X\|_{\psi_2} = \inf\{K > 0 : \mathbb{E}[e^{X^2/K^2}] \le 2\}$. It is shown in Vershynin (2018, Section 2.6.1) that, if $X$ follows a centered Gaussian distribution with standard deviation $\sigma$, then $\|X\|_{\psi_2} = \sigma\sqrt{8/3}$ and $\|X\|_{L_2} = \sigma$. Hence, $K = \sqrt{8/3}$ in our case and we have

$$-\frac{8C}{3}\Big(\sqrt{\frac{D+u}{n}} + \frac{D+u}{n}\Big)I_D \preceq \Sigma^{-1/2}\hat{\Sigma}\Sigma^{-1/2} - I_d \preceq \frac{8C}{3}\Big(\sqrt{\frac{D+u}{n}} + \frac{D+u}{n}\Big)I_D.$$

By multiplying $\Sigma^{1/2}$ from left and right for both side, we derive that

$$-\frac{8C}{3}\Big(\sqrt{\frac{D+u}{n}} + \frac{D+u}{n}\Big)\Sigma \preceq \hat{\Sigma} - \Sigma \preceq \frac{8C}{3}\Big(\sqrt{\frac{D+u}{n}} + \frac{D+u}{n}\Big)\Sigma.$$

$\square$

# D. Useful Lemma

In this section we provide some lemma that will be useful throughout the whole paper (see also, Horn & Johnson, 2012, Section 7.7).

**Lemma D.1.** *Let $A, B$ be two symmetric $D \times D$ real matrices, and $S$ be an arbitrary $D \times D$ real matrix. The following statements hold:*

  *(i) If $A \preceq B$, then $S^\top A S \preceq S^\top B S$.*

  *(ii) If $A^2 \preceq B^2$, then $A \preceq B$. In particular if $A$ and $B$ are semi-definite positive, then $A \preceq B \Rightarrow \sqrt{A} \preceq \sqrt{B}$.*

# E. Experiment details

## E.1. Synthetic data experiment

**Synthetic Gaussian data.** In the experiment of Figure 3, we generate data using Gaussian distribution with covariance matrices equal to $\text{diag}(1, 0.6, 0.6^2, \ldots, 0.6^6, 10^{-10}, 10^{-10})$ (left) and $\text{diag}(10, 0.2, 0.2, 0.2, 0, 0)$ (right). We then generate sample by first estimating the variances with the data with 1k sample, then solving the SDE (11) separately for each projection. We generate new sample using the Ornstein-Uhlenbeck process with $T = 2$ and 1000 discretization steps.

**Synthetic Gaussian data with exponentially decaying eigenvalues.** In the experiment of Figure 4, we generate data using Gaussian distribution with covariance matrices equal to $\text{diag}(1, 1/4, 1/4^2, \ldots, 1/4^9)$. We then train a linear model with score matching using 10 thousand samples, and we clip the parameters of the model. We generate new sample using the Ornstein-Uhlenbeck process with $T = 2$ and 1000 discretization steps.

## E.2. Natural image experiment

**Common details.** We use the dataset CelebA and CelebA-HQ (Liu et al., 2015) and ImageNet-256 (Deng et al., 2009). We use a U-Net model (Ronneberger et al., 2015) and an Adam optimizer (Kingma, 2014). The diffusion model uses variance preserving noise schedule (Song et al., 2020). The code was implemented in JAX (Bradbury et al., 2018).

**Training of AE on CelebA-HQ.** We train a VQ-VAE using the VQ-GAN loss (Esser et al., 2021) for 1.95 million step on 20 TPUv2. The VQ-VAE encodes the images to a latent space of shape $64 \times 64 \times 3$ and reaches a 2k-rFID score of 2.44. Other hyperparameters for training is summarized in Table 1.

| Name | Value |
|---|---|
| Coefficient of the adversarial loss | 0.1 |
| Coefficient of the generator loss | 100 |
| Coefficient of the LPIPS loss | 1.0 |
| Coefficient of the discriminator loss | 0.01 |
| Number of embeddings of the vector quantizer | 8192 |
| Optimizer | Adam with standard hyperparameters |
| EMA decay | 0.9999 |
| Learning rate | $10^{-5}$ |
| Batch size | 16 |

*Table 1.* Hyperparameters for training VQ-VAE on CelebA-HQ.

**Training of AE on ImageNet-256.** We train a VQ-VAE using the VQ-GAN loss (Esser et al., 2021) for 715 thousand (resp. 1.81 million) steps on 20 TPUv2 for a latent shape of $32 \times 32 \times 4$ (resp. $64 \times 64 \times 3$). The VQ-VAE achieves an rFID score of 3.6 (resp. 4.2). Other hyperparameters for training is summarized in Table 3.

**Training of LDM on CelebA-HQ.** We train an LDM on the images encoded by the AE we described above. We train for 5.25 million steps on 8 TPUv6. We summarize the hyperparameters used in Table 2.

| Name | Value |
| --- | --- |
| Noise schedule | Variance Preserving |
| Number of sampling steps | 250 |
| Optimizer | Adam with standard hyperparameters |
| EMA decay | 0.9999 |
| Learning rate | $10^{-4}$ |
| Batch size | 16 |

*Table 2.* Hyperparameters for training LDM on encoded images of CelebA-HQ.

| Name | Value |
| --- | --- |
| Coefficient of the adversarial loss | 0.1 |
| Coefficient of the generator loss | 100 |
| Coefficient of the LPIPS loss | 1.0 |
| Coefficient of the discriminator loss | 0.01 |
| Number of embeddings of the vector quantizer | 8192 |
| Optimizer | Adam with standard hyperparameters |
| EMA decay | 0.999 |
| Learning rate | $10^{-5}$ |
| Batch size | 16 |

*Table 3.* Hyperparameters for training VQ-VAE on ImageNet-256.

**Training of LDM on ImageNet-256.**  We train two LDMs on the images encoded by the AEs we described above. We train for 1 million steps on 8 TPUv6 for both LDMs. We summarize the hyperparameters used in Table 5.

**Training pixel diffusion model on CelebA and ImageNet-64.**  We train diffusion models directly on CelebA and ImageNet-64. We train both models for 1 million steps on 12 TPUv2. We summarize the hyperparameters in Table 4.

**Results.**  We previously introduced some results in Section 1. Here, we present additional evidence regarding the quality of the generated images. We observe (Figure 14) that in the final few steps, the sample of LDM does not change visibly. On the contrary, the images generated in pixel space (Figure 15) are still denoised even in the last steps.

**Linear AE and U-Net LDM on MNIST (Deng, 2012).**  We train linear AE on MNIST with dimension 64, 256, and 400 for 5k steps each using 4 TPUv2. Then we train U-Net diffusion models paired with each AE for 10k steps each using 4 TPUv2. We summarize other hyperparameters in Table 6 and 7.

| Name | Value |
|---|---|
| Noise schedule | Variance Preserving |
| Number of sampling steps | 250 |
| Optimizer | Adam with standard hyperparameters |
| EMA decay | 0.9999 |
| Learning rate | $10^{-4}$ |
| Batch size | 128 |

*Table 4.* Hyperparameters for training diffusion model on CelebA and ImageNet-64.

| Name | Value |
|---|---|
| Noise schedule | Variance Preserving |
| Number of sampling steps | 250 |
| Optimizer | Adam with standard hyperparameters |
| EMA decay | 0.999 |
| Learning rate | $10^{-4}$ |
| Batch size | 1024 |

*Table 5.* Hyperparameters for training LDM on encoded images of ImageNet-256.

| Name | Value |
|---|---|
| Optimizer | Adam with standard hyperparameters |
| Learning rate | 0.003 |
| Batch size | 256 |

*Table 6.* Hyperparameters for training Linear AE on MNIST.

| Name | Value |
|---|---|
| Noise schedule | Variance Preserving |
| Number of sampling steps | 250 |
| Optimizer | Adam with standard hyperparameters |
| EMA decay | 0.999 |
| Learning rate | $10^{-4}$ |
| Batch size | 256 |

*Table 7.* Hyperparameters for training LDM on encoded images of MNIST.

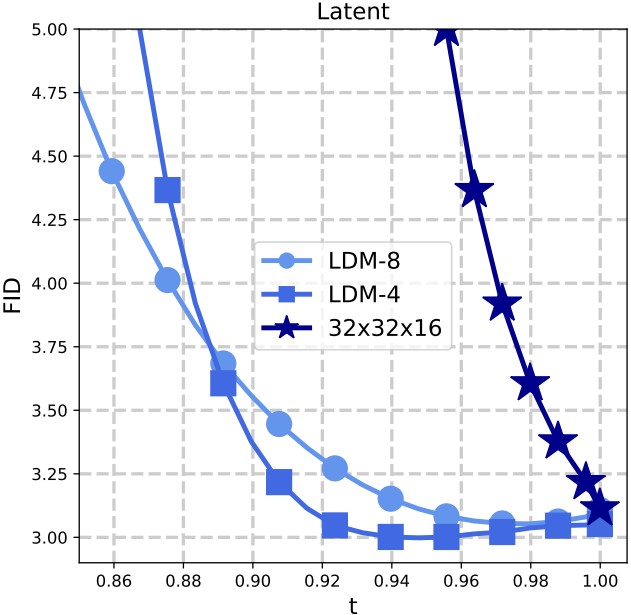

*Figure 11.* Zoom in of Figure 7.

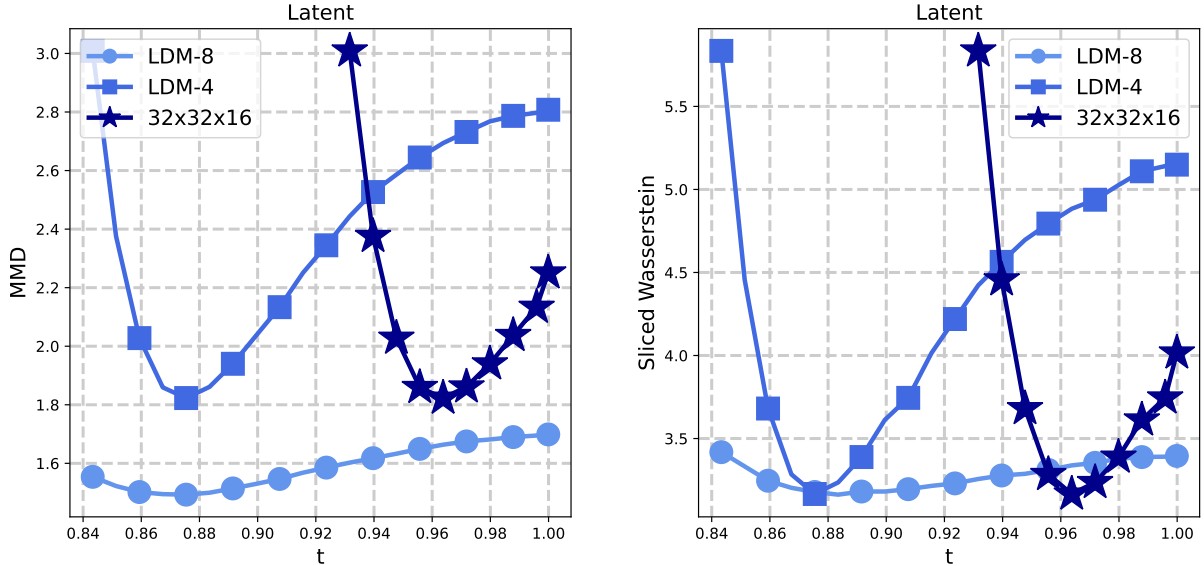

*Figure 12.* We measure the image quality of different LDMs on ImageNet-256 by MMD and Sliced Wasserstein distance.

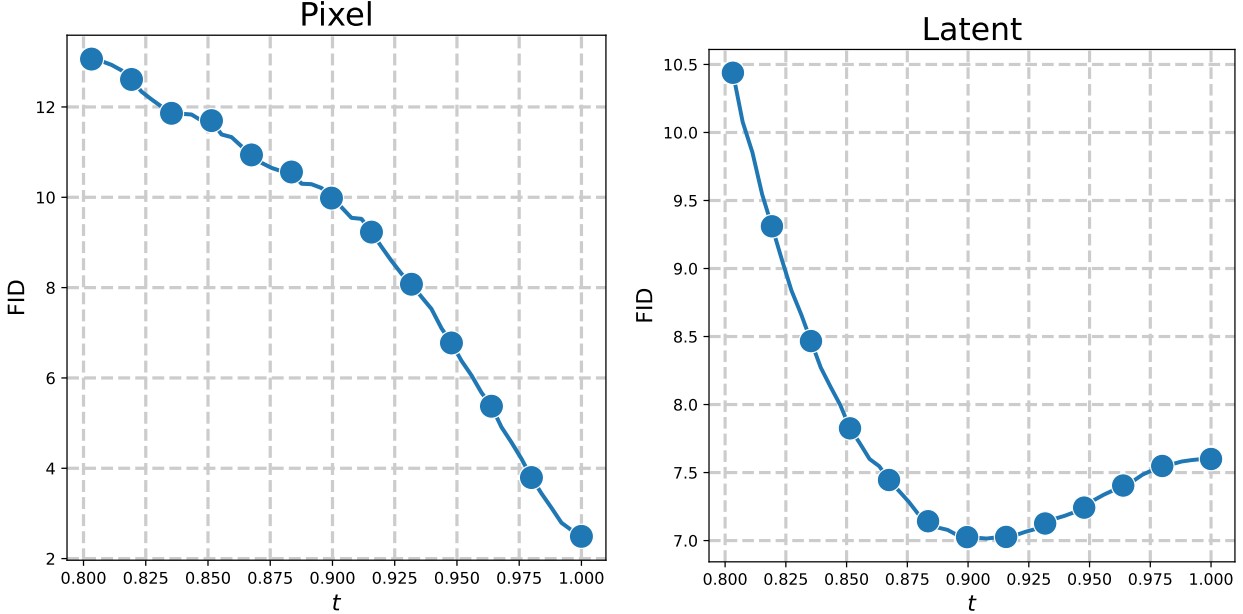

*Figure 13.* We train AE on CelebA-HQ and a corresponding LDM. In addition, we compare it to the FID curve of pixel diffusion model trained on CelebA. The experiment shows monotonicity in FID curve of pixel diffusion, while the FID curve of LDM achieves a minimum before the final diffusion time.

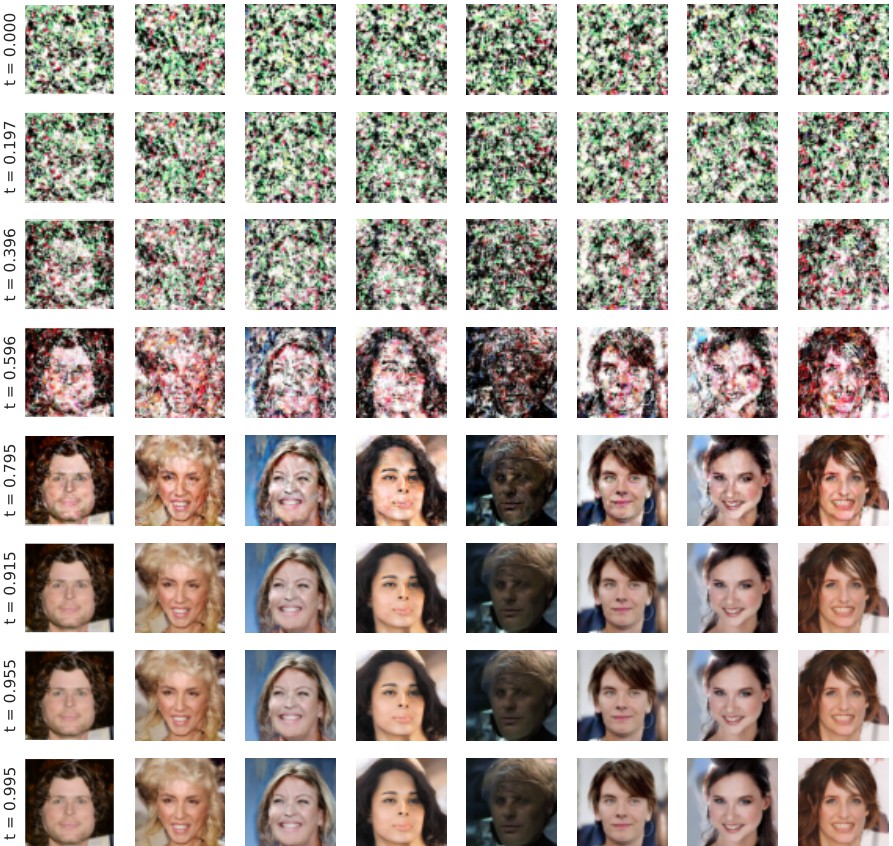

*Figure 14.* The final steps of LDM do not improve image quality.

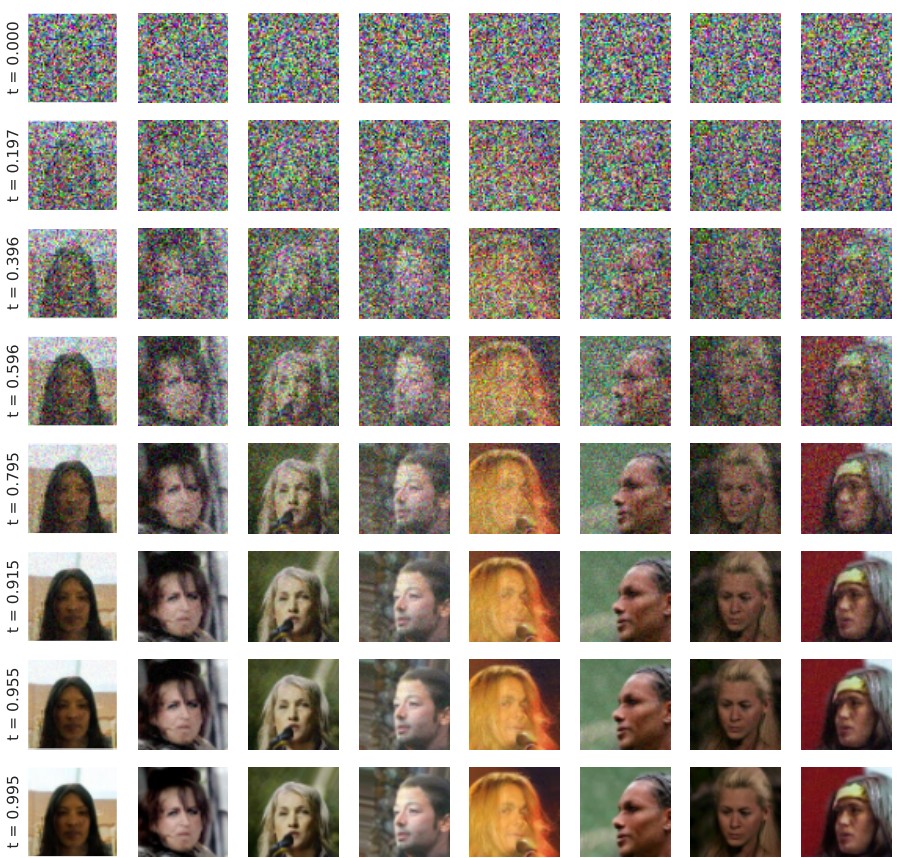

*Figure 15.* The quality of sample in diffusion on pixel space is still increasing in the final few steps.

