# OpenReview forum: "Optimal Stopping in Latent Diffusion Models"
_ICML.cc/2026/Conference — ICML 2026 regular_

### Official Review · Reviewer_woJ6 · 2026-03-03

**Soundness:** 3
**Presentation:** 3
**Significance:** 3
**Originality:** 3
**Overall Recommendation:** 4
**Confidence:** 3

**Summary:**

This paper investigates the optimal stopping time in latent diffusion models and its interaction with latent space dimensionality and other hyperparameters. Under the assumption of Gaussian data and a linear autoencoder, the authors establish in Proposition 4.1 that generation quality is not monotonic with respect to the stopping time. Moreover, given a stopping time $t$, Proposition 4.2 provides the optimal latent space dimension. Numerical experiments on synthetic Gaussian data and ImageNet-256 validate the theoretical findings.

**Compliance With Llm Reviewing Policy:**

Affirmed.

**Final Justification:**

This paper investigates the optimal stopping time in latent diffusion models and its interaction with latent space dimensionality and other hyperparameters. The authors address my main concerns in the rebuttal. Thus, I maintain my score of 4.

**Key Questions For Authors:**

- How could the proposed analysis be extended to more general settings, such as nonlinear autoencoders or data sampled from a Gaussian mixture model?
- Is there an explicit criterion for selecting the optimal stopping time? Proposition 4.1 only establishes that the evaluation metric is non-monotonic in time, but does not provide a concrete guideline for choosing the optimal $t$.
- Several FID scores reported in the bottom panel of Figure 4 are smaller than $1$. Is this expected under the experimental setup?

**Limitations:**

Yes.

**Strengths And Weaknesses:**

Strengths:
- Provides clear theoretical results on the relationship between early stopping time and latent dimensionality in latent diffusion models under well-defined settings.
- Numerical experiments validate the theoretical findings.

Weaknesses:
- Soundness:
  - The theoretical analysis is conducted under the assumption of Gaussian data and a linear autoencoder, which differ substantially from the complex data distributions and architectures used in practice. Although experiments on ImageNet-256 are included, the generalizability of the simplified theoretical framework may be limited.
- Presentation:
  - There are inconsistencies in punctuation usage; for instance, several sentences—such as in Proposition 4.2—are missing a concluding period.

---

> ### Author Rebuttal · Authors · 2026-03-31
>
> > How could the proposed analysis be extended to more general settings, such as nonlinear autoencoders or data sampled from a Gaussian mixture model?
>
> We thank the reviewer for raising this question concerning generalizability of our theoretical framework.
>  - Theoretically, extending our results to GMM settings requires analysis of empirical estimation of GMM and exact bound for Wasserstein distance between mixture distributions. While an exact closed-form solution is analytically challenging, the fundamental mechanism holds. To verify this, we conducted new experiments on GMM data (see response to Reviewer BAmE, Q3). The new results confirm that our theoretical findings hold in the more complex setting: the optimal projection dimension is exactly the intrinsic dimension of the data, and the U-shape of the distance between generated and true distribution is clearly observable.
>  - To extend the analysis to non-linear AE would require a local approach, analyzing local linearization of the decoder on the data manifold. Nonetheless, we provided strong empirical findings that such extension is valid in section 6, which we find strong alignments in noisy AE and LDM in our experiments, which use highly non-linear AE on ImageNet-$256$.
>
> > Is there an explicit criterion for selecting the optimal stopping time? Proposition 4.1 only establishes that the evaluation metric is non-monotonic in time, but does not provide a concrete guideline for choosing the optimal $t$.
>
> In our theoretical framework, the optimal stopping time boils down to solving a non-trivial equation. In section 6, we reveal a practical guideline of selecting such stopping time, by stopping at the minimum provided by  the noisy AE proxy.
>
> > Several FID scores reported in the bottom panel of Figure 4 are smaller than $1$. Is this expected under the experimental setup?
>
> The FID score is smaller than $1$ since we are reporting the reconstructed FID score of the AE, where the generated samples are the encoded then decoded images by the AE.

---

> > ### Author Rebuttal · Reviewer_woJ6 · 2026-04-04
> >
> > I thank the authors for the response. They address all the previous questions. However, I have an additional question:
> > - The reverse process of diffusion models produce marginal distribution ($q_{t} (x)$) exactly the same as that in the forward process ($p_{t} (x)$), i.e., $q_{t} (x) = p_{t} (x)$, while for $t > 0$, the marginal $p_{t} (x)$ is a noisy distribution perturbed by the actual data distribution $p_{0} (x)$, which indicates $p_{t} (x) \neq p_{0} (x)$. Despite that the empirical results show that early stop improves the generation quality, this is not the case from a theoretical perspective. Therefore, could the authors shed more light on this gap between theory and practice?
> >
> > I maintain my positive score.

---

> > > ### Author Response · Authors · 2026-04-04
> > >
> > > We thank the reviewer for raising this fundamental question. The SDE matching only guarantees distribution equality in the space where the diffusion operates (i.e., $q_0(E(x))=p_0(E(x))$), and if the score is learned perfectly. In this setting, we agree that running the diffusion until the end is optimal. In the case of LDMs, lossy reconstruction from the latent space as well as errors in the score estimation induce a mismatch between the reconstructed distribution $D(q_0(E(x))$ and $p_0(x)$. Our contribution is to characterize how this mismatch leads to a benefit from early stopping, both theoretically in the Gaussian case and experimentally through GMM ablation study and real-world image generation tasks. Hence we do not see a gap between theory and practice in this regard. We will clarify this point in the next version.

---

### Official Review · Reviewer_QeyP · 2026-03-12

**Soundness:** 3
**Presentation:** 3
**Significance:** 2
**Originality:** 3
**Overall Recommendation:** 4
**Confidence:** 4

**Summary:**

This paper studies the question of how the dimension of the latent space for a latent diffusion model impacts the optimal time at which to stop the reverse process. Whereas for pixel space diffusion models, the quality of the generated sample progressively improves until the very end, for latent diffusion models the quality plateaus at some point before the end, and this paper develops some theory in an exactly solvable Gaussian setting to explain this finding. In a toy model where the true data is Gaussian and the estimated score corresponds to a Gaussian, they observe that if these Gaussians are different, the distance between the generated distribution and the true one is not monotonically decreasing over the course of the reverse process, and furthermore this effect is quantitatively more pronounced at lower latent dimensions. They corroborate these findings with experimental results on ImageNet, and observe that the optimal stopping time can be reliably predicted by a cheaper proxy model in which one simply encodes a sample, and decodes it after applying varying levels of noise.

**Compliance With Llm Reviewing Policy:**

Affirmed.

**Final Justification:**

The rebuttal addressed my concerns. While I remain positive about the direction of this work, I will keep my score at 4 primarily because the gains from stopping optimally in the experiments are somewhat marginal.

**Key Questions For Authors:**

- For the noisy AE experiments, in $D(b_t E(x) + a_t Z)$, is $x$ supposed to just be a sample from the dataset?
- If one trained a *linear* score network on ImageNet, and also computed the empirical mean and covariance of ImageNet, this would give heuristic values that one could in principle plug into the formulas computed in Section 4. I'm curious how predictive the results would be.

**Limitations:**

The paper doesn't explicitly address limitations and could benefit from a discussion around whether the U-curve relationship between quality and stopping time is sufficiently significant to merit carefully tuning the stopping time.

**Strengths And Weaknesses:**

**Strengths**
- They comprehensively investigate the case of Gaussian data and score estimates and give exact characterizations for the optimal stopping time and for the optimal latent dimension.
- The noisy AE appears to be generally predictive of when FID stops improving along the reverse process.
- The paper gives a conceptually satisfying message for why pixel diffusion and latent diffusion differ with regards to the role of early stopping, and how latent dimension affects optimal stopping tim.

**Weaknesses**
- One obvious complaint is that the theory is very specific to Gaussian data and linear score estimates, though this is arguably not so bad as the role of this theory is mainly to provide a clarifying conceptual picture for what appears to be a real phenomenon.
- The noisy AE isn't fully predictive, e.g. the intersection of the curves for d = D/12 and d = D/48 do not agree between LDM and AE in Figure 4.
- The U-curve behavior in Figure 4 is barely perceptible, and it's unclear from the qualitative examples given in the paper whether simply running the reverse process until the end would really hurt the quality of the samples generated

---

> ### Author Rebuttal · Authors · 2026-03-31
>
> > One obvious complaint is that the theory is very specific to Gaussian data and linear score estimates, though this is arguably not so bad as the role of this theory is mainly to provide a clarifying conceptual picture for what appears to be a real phenomenon.
>
>  We thank the reviewer for this positive remark. A more detailed discussion can be found in our responses to the other reviewers, where we elaborate further on this point (see answer to Reviewer BAmE for new experiments with mixtures of Gaussian).
>
> > The noisy AE isn't fully predictive, e.g. the intersection of the curves for $d = D/12$ and $d = D/48$ do not agree between LDM and AE in Figure 4.
>
> The reviewer is right that there is a numerical deviation in the exact intersection points between the Noisy AE and the full LDM in Figure 4. This is expected since the Noisy AE is a highly efficient heuristic, not a perfect 1:1 simulation. However, the noisy AE provides practical value by allowing relative performance rankings across different latent dimensions and an approximate interval for the optimal stopping time of diffusion.
>
> > The U-curve behavior in Figure 4 is barely perceptible, and it's unclear from the qualitative examples given in the paper whether simply running the reverse process until the end would really hurt the quality of the samples generated
>
> We acknowledge that the U-curve in Figure 4 is subtle. However, when analyzing the zoomed-in trajectories Figure 5 (FID), as well as Figure 8 (MMD and Sliced Wasserstein), the quantitative degradation is consistent. Regarding the qualitative visual impact (Figure 12),  we agree that the difference is not visually flagrant (although there are slight texture differences). Because the qualitative difference is marginal but the quantitative metrics degrade, running the reverse process to the end is computationally wasteful at best and mathematically suboptimal at worst.
>
> > For the noisy AE experiments, in $D(b_tE(x)+a_t Z)$, is $x$ supposed to just be a sample from the dataset?
>
> Thank you for this very helpful remark. We agree that this point was not clearly specified: $x$ should be sampled from the dataset and $Z$ is sampled from standard Gaussian noise. We will clarify this in the next version of the paper.
>
> > If one trained a linear score network on ImageNet, and also computed the empirical mean and covariance of ImageNet, this would give heuristic values that one could in principle plug into the formulas computed in Section 4. I'm curious how predictive the results would be.
>
> Our theoretical formulas in Section 4 are designed to provide cheap heuristic prediction for optimal stopping and dimension reduction. While computing and estimating the full empirical covariance matrix of ImageNet latents (e.g. a $12288 \times 12288$ matrix for a latent space $d=64\times64\times3$) and plugging it into a linear score network is an interesting experiment,  the Noisy AE we presented in Section 6 is actually the practical, non-linear realization of this exact heuristic. Instead of a massive linear computation, the Noisy AE directly simulates the noise injection within the non-linear latent space, achieving the highly predictive results you hypothesized. Additionally, we conducted new experiments on mixtures of Gaussian (cf. the answer to Q3 of Reviewer BAmE), and we show that our findings are still observed in the GMM setting.

---

> > ### Author Rebuttal · Reviewer_QeyP · 2026-04-03
> >
> > I thank the authors for their careful response. In general it answers the remaining questions I had about the work, though the point about "running the reverse process to the end is computationally wasteful at best and mathematically suboptimal at worst" is still slightly unsatisfactory --- running the reverse process to the end without carefully tuning the point of early stopping could potentially be more practical than expending computational resources to determine when to early stop. Perhaps the authors' point could be further strengthened by trying out other modalities beyond images, where the difference could be more pronounced? In general I maintain my positive opinion about this work and keep my score.

---

> > > ### Author Response · Authors · 2026-04-03
> > >
> > > We thank the reviewer for the positive assessment. We highlight that our proposed noisy AE proxy is a nearly free method for determining the optimal stopping time. We circumvent the need for costly evaluation by decoding noise-injected latents. In addition, it is known in the literature (cf. [1][2]) that it is necessary to have denser timesteps when $t\to0$ for better convergence, which makes running diffusion to the end more computationally costly. We agree further experiments on other modalities are interesting. For example, in video generation where the decoder is sensitive to temporal consistency and high-frequency noise, we suspect that the benefit of early stopping will be more pronounced.
> > >
> > >
> > > [1] Benton, Joe, et al. "Nearly $ d $-linear convergence bounds for diffusion models via stochastic localization." arXiv:2308.03686 (2023).
> > >
> > > [2] Li, Gen, and Yuling Yan. "O (d/T) convergence theory for diffusion probabilistic models under minimal assumptions." Journal of Machine Learning Research 26.292 (2025): 1-55.

---

### Official Review · Reviewer_FWyb · 2026-03-15

**Soundness:** 3
**Presentation:** 3
**Significance:** 2
**Originality:** 3
**Overall Recommendation:** 4
**Confidence:** 3

**Summary:**

This paper investigates the problem of optimal stopping time in latent diffusion models. It shows that generation quality does not necessarily reach its optimum at the final denoising step, but instead depends on a trade-off between latent dimensionality and the noise removal process. Under a Gaussian framework, the authors derive the distance between the generated distribution and the true data distribution, theoretically demonstrating that early stopping can improve generation quality. The authors further propose using a noisy autoencoder as a proxy model to predict diffusion model performance, enabling a faster selection of latent dimensions and stopping time.

**Compliance With Llm Reviewing Policy:**

Affirmed.

**Key Questions For Authors:**

none

**Limitations:**

No limitations are mentioned in the paper.

**Strengths And Weaknesses:**

## Strength

1. Provided a rigorous and convincing theoretical framework to explain the early-stopping phenomenon in Latent Diffusion.

2. The presentation of the whole progressive derivation is good and easy to follow.

3. The experimental results in the real-world datasets confirm the theory (derived from the Gaussian framework).

## Weakness

1. Section 6 (Evaluation) is overall too short and looks simple.

2. I suggest that the authors can add empirical results using out-of-the-box SD-VAE and Flux-AE, to verify the phenomenon that 'the final steps of the diffusion can degrade sample quality in latent diffusion models. Which, in turn, can improve the importance and generalization of this paper.

3. I believe the findings and methods (latent perturbation) in [1][2][3] are related to this paper, and can be discussed in the paper.

4. The term 'VQ-VAE' is wrongly used in the paper; we normally refer to it as KL-VAE since VQ-VAE is the vector-quantized one used for autoregressive generation.


[1] Input perturbation reduces exposure bias in diffusion models. ICML, 2023.

[2] Nextstep-1: Toward autoregressive image generation with continuous tokens at scale. arXiv preprint arXiv:2508.10711

[3] Diffusion transformers with representation autoencoders. ICLR, 2026.

---

> ### Author Rebuttal · Authors · 2026-03-31
>
> > 1. Section 6 (Evaluation) is overall too short and looks simple.
>
> > 2. I suggest that the authors can add empirical results using out-of-the-box SD-VAE and Flux-AE, to verify the phenomenon that 'the final steps of the diffusion can degrade sample quality in latent diffusion models. Which, in turn, can improve the importance and generalization of this paper.
>
> We appreciate the reviewer's push to expand our empirical evaluation. To address this and verify the phenomenon on widely used models, we conducted new experiments using SD-VAE and Flux-AE, as suggested. These new results confirm that the U-shape degradation in FID (and Sliced Wasserstein) persists even in these out-of-the-box AEs. Furthermore, the optimal stopping time of LDM is aligned again with the minimum of the noisy AE.  We will report these new results in the next version of the paper. We would be glad to discuss if the reviewer has any further comments or suggestions to further strengthen our empirical evaluation.
>
> > 3. I believe the findings and methods (latent perturbation) in [1][2][3] are related to this paper, and can be discussed in the paper.
>
> We appreciate the suggested references that we will incorporate in the revised version.
>
> > 4. The term 'VQ-VAE' is wrongly used in the paper; we normally refer to it as KL-VAE since VQ-VAE is the vector-quantized one used for autoregressive generation.
>
> Thank you for pointing this out. We will revise the terminology accordingly and replace “VQ-VAE” with “KL-VAE” where appropriate in the paper.

---

> > ### Author Rebuttal · Reviewer_FWyb · 2026-04-02
> >
> > Thanks to the authors for addressing my concerns.
> >
> > If you already have the experimental results using SD-VAE and Flux-AE, please feel free to add in the rebuttal

---

> > > ### Author Response · Authors · 2026-04-04
> > >
> > > The results of additional experiments can be found here ([link](https://anonymous.4open.science/r/ICML_rebuttal_optimal_stopping_ldm)). The new results confirm the U-shape degradation in these off-the-shelf AEs. Furthermore, we provide results on LDM paired with SD-VAE, which show that the optimal stopping time of the LDM aligns with the minimum of the Noisy AE proxy. Given the computational constraints of the short rebuttal window, completing a full LDM training run from scratch on Flux-AE was not feasible. However, this highlights the utility of our proposed heuristic: the trajectory of noisy AE (that requires zero training) as a proxy for optimal stopping for LDM on Flux-AE. For additional details of the experiment, we refer to README.md in the link.

---

### Official Review · Reviewer_BAmE · 2026-03-18

**Soundness:** 2
**Presentation:** 4
**Significance:** 3
**Originality:** 3
**Overall Recommendation:** 3
**Confidence:** 3

**Summary:**

This paper studies a phenomenon in latent diffusion models where running the sampler all the way to the final denoising step can reduce generation quality, so stopping early can work better. The authors analyze this in a Gaussian and linear autoencoder setting and show that the best stopping point depends strongly on the latent dimension, with smaller latent spaces often preferring earlier stopping. They also suggest using noisy autoencoders as a cheap way to choose the latent dimension and stopping point, and support the idea with experiments on synthetic data and image generation benchmarks.

**Compliance With Llm Reviewing Policy:**

Affirmed.

**Final Justification:**

The authors addressed most of my questions during the rebuttal, and I have accordingly raised my score to a 3. However, my main concern remains that the core analysis relies on highly restrictive assumptions, including Gaussian data and linear autoencoders. While the proposed hypothesis is interesting and potentially important, I am still not convinced that the explanation carries over to realistic latent diffusion models without stronger empirical or theoretical justification.

**Key Questions For Authors:**

1) How far do you believe the main theoretical explanation extends beyond the Gaussian and linear autoencoder setting studied in the paper? While Section 6 demonstrates empirical alignment, what evidence supports that the same mathematical mechanism drives early stopping in realistic nonlinear LDMs, rather than this being mainly a property of the simplified model?

2) Why is Wasserstein-2 distance the right object for the theory, given that the motivating failure mode is high-frequency decoder induced pixel artifacts? It is not obvious to me that this metric captures the kind of degradation the paper is trying to explain.

3) How does the paper empirically distinguish the intrinsic dimensionality explanation from alternative explanations of early stopping, such as numerical instability or poor score estimation near the final steps (despite the theoretical efforts to isolate them in Section 5)? At present, it is not clear to me whether the proposed explanation is the main cause of the phenomenon in practice.

**Limitations:**

Yes

**Strengths And Weaknesses:**

Soundness: The paper is sound within its intended setting, and the theory is supported by experiments showing that noisy autoencoders track LDM stopping behavior and that larger latent spaces tend to prefer later stopping times. My main concern is that the central analysis relies on very restrictive assumptions, including Gaussian data, linear autoencoders, and in Section 5 a restricted diagonal score class with bounded norm, so it is unclear how much of the explanation carries over to realistic LDMs. I am also not fully convinced by the use of Wasserstein-2 distance in the theory, since the motivating claim is about decoder induced pixel artifacts, which this metric may not capture well. Finally, while the paper argues for an intrinsic dimensionality based explanation of early stopping, it does not fully disentangle this from alternative explanations such as numerical instability or poor score estimation near the final steps.

Presentation: The paper is clearly written and well structured.

Significance: The paper studies an important practical question in latent diffusion models, namely why early stopping can improve sampling quality.

Originality: The main novelty is a new theoretical explanation for why the best stopping time depends on latent dimension, together with the idea of using a noisy autoencoder as a cheap stand in for the full model.

---

> ### Author Rebuttal · Authors · 2026-03-31
>
> > 1. How far do you believe the main theoretical explanation extends beyond the Gaussian and linear autoencoder setting studied in the paper? While Section 6 demonstrates empirical alignment, what evidence supports that the same mathematical mechanism drives early stopping in realistic nonlinear LDMs, rather than this being mainly a property of the simplified model?
>
> We thank the reviewer for raising this question. We emphasize that an important contribution of the paper is providing a practical and actionable guideline to select the optimal stopping time and latent dimension in practice. In addition, We strongly agree with Reviewer QeyP's assessment that the role of our exact theoretical setting is "mainly to provide a clarifying conceptual picture for what appears to be a real phenomenon." Furthermore, in section 6, experiments concerning Noisy AE provide strong evidence that bridges the theoretical framework and practice as follows. We evaluated a "noisy AE" proxy using complex U-Net-based AEs on ImageNet-$256$ to test the noise-injection mechanism. As shown in Figure 4, the FID trajectories of the noisy AE strictly align with those of the full LDM. In consequence, the noisy AE is a good proxy to predict  both the optimal stopping times and the intersection points of performance across different latent dimensions ($d = D/48, D/16, D/12$).
>
> > 2. Why is Wasserstein-2 distance the right object for the theory, given that the motivating failure mode is high-frequency decoder induced pixel artifacts? It is not obvious to me that this metric captures the kind of degradation the paper is trying to explain.
>
> We chose Fréchet distance (which is equivalent to Wasserstein distance between 2 Gaussian distributions) since this is the mathematical formulation of the FID score, which is the standard metric used for image generation evaluation. The features captured by the Inception-v3 network are sensitive to the disruption in high-frequency artifacts. To ensure robustness, we also reported Maximum Mean Discrepancy (MMD) and Sliced Wasserstein distance. We remark that both metrics reaffirm the U-shaped as observed for FID score (see Figure 8).
>
> > 3. How does the paper empirically distinguish the intrinsic dimensionality explanation from alternative explanations of early stopping, such as numerical instability or poor score estimation near the final steps (despite the theoretical efforts to isolate them in Section 5)? At present, it is not clear to me whether the proposed explanation is the main cause of the phenomenon in practice.
>
> Thank you for suggesting this very interesting perspective, which leads us to conduct subsequent experiments. As an ablation study, we use a synthetic mixture of Gaussian data and use PCA for dimension reduction. In this setting, we can compute the score directly, which eliminates the score estimation error. In addition, we use very dense timestep discretization and score clipping to minimize numerical instability. The U-shape persists even with perfect scores and negligible discretization error. In addition, the optimal latent dimension is exactly the same as the intrinsic dimension of the data distribution. We will add these experiments to the next version.

---

> > ### Author Rebuttal · Reviewer_BAmE · 2026-04-02
> >
> > Thank you for your response. Would the authors be able to answer the following questions?
> >
> > Q1: Would you mind explaining in a bit more detail the following: "Furthermore, in section 6, experiments concerning Noisy AE provide strong evidence that bridges the theoretical framework and practice as follows. We evaluated a "noisy AE" proxy using complex U-Net-based AEs on ImageNet-$256$ to test the noise-injection mechanism. As shown in Figure 4, the FID trajectories of the noisy AE strictly align with those of the full LDM. In consequence, the noisy AE is a good proxy to predict both the optimal stopping times and the intersection points of performance across different latent dimensions ($d = D/48, D/16, D/12$)."
> >
> > In particular, does this address my question regarding "what evidence supports that the same mathematical mechanism drives early stopping in realistic nonlinear LDMs, rather than this being mainly a property of the simplified model"?
> >
> > Q2: Resolved.
> >
> > Q3: Would the authors mind providing more details regarding "we use very dense timestep discretization and score clipping to minimize numerical instability"?

---

> > > ### Author Response · Authors · 2026-04-03
> > >
> > > > Q1: Would you mind explaining in a bit more detail the following: "Furthermore, in section 6, experiments concerning Noisy AE provide strong evidence that bridges the theoretical framework and practice as follows. We evaluated a "noisy AE" proxy using complex U-Net-based AEs on ImageNet-$256$ to test the noise-injection mechanism. As shown in Figure 4, the FID trajectories of the noisy AE strictly align with those of the full LDM. In consequence, the noisy AE is a good proxy to predict both the optimal stopping times and the intersection points of performance across different latent dimensions ($d = D/48, D/16, D/12$)."
> > >
> > > > In particular, does this address my question regarding "what evidence supports that the same mathematical mechanism drives early stopping in realistic nonlinear LDMs, rather than this being mainly a property of the simplified model"?
> > >
> > > In our Gaussian framework (Section 4), we show that the structural mechanism of LDM (dimensionality reduction coupled with noise due to score estimation) produces a U-shape trade-off. The reviewer rightly asks if this trade-off also drives early stopping in realistic nonlinear LDMs. To prove this we utilize the noisy AE proxy, where we decode noise-injected latents. In particular, the noisy AE completely bypasses the score matching, and it solely tests the decoder's reaction to latent noise. As shown in Figure 4, the full LDM degrades in the same manner as the noisy AE proxy. Because the full LDM strongly aligns with the proxy, it supports our theoretical bridge: the early-stopping phenomenon in practice is governed by the structural mechanism we identified in theory.
> > >
> > > > Q3: Would the authors mind providing more details regarding "we use very dense timestep discretization and score clipping to minimize numerical instability"?
> > >
> > > To isolate our proposed mechanism from numerical instability, we ensure the SDE solver is as stable as possible for the GMM ablation study. By using a very high number of discretization steps (5000 steps) when solving the reverse process, we ensure the numerical trajectory closely approximates the true continuous-time SDE. Furthermore, to prevent the score from blowing up when $t\to0$ (cf. [1]), we explicitly bound the magnitude of the score (as we did in our analysis in Section 5). Since the U-shape degradation and optimal latent projection persist while ensuring numerical stability, we conclude it is driven by the latent dimensionality.
> > >
> > > [1] Yang, Zhantao, et al. "Lipschitz singularities in diffusion models." The Twelfth International Conference on Learning Representations. 2023.

---

### Decision · Program_Chairs · 2026-04-30

**Decision:**

Accept (regular)

**Comment:**

This paper investigates the optimal stopping time in latent diffusion models and its interaction with latent dimensionality.
Under a Gaussian data and linear autoencoder setting, it develops a clear theoretical explanation on why early stopping can improve generation quality, and supports these insights with both synthetic and real-data experiments.

The reviewers' opinions are **mixed but overall leaning towards acceptance**.
They agree that the paper provides interesting and valuable theoretical insights into an important and previously underexplored phenomenon.
At the same time, they note that the current analysis relies on simplified assumptions and that the empirical validation remains somewhat limited in scope and depth.

Taking these points together, I find that the paper's theoretical contribution is sufficiently interesting and meaningful to warrant acceptance, despite its current limitations (e.g., holds for Gaussian data and linear autoencoders).
I therefore recommend acceptance.
For the final version, **I strongly encourage the authors to strengthen the empirical evaluation and broaden the experimental validation**, in line with the reviewers' suggestions, to further enhance the impact of the work.